# Soil moisture and streamflow deficit anomaly index: An approach to quantify drought hazards by combining deficit and anomaly

Eklavyya Popat[1], Petra Döll[1,2]

[1]Institute of Physical Geography, Goethe University Frankfurt, Germany
[2]Senckenberg Leibniz Biodiversity and Climate Research Centre Frankfurt (SBiK-F), Frankfurt, Germany

*Correspondence to*: Eklavyya Popat (popat@em.uni-frankfurt.de)

**Abstract**

Drought is understood as both a lack of water (i.e., a deficit as compared to demand) and a temporal anomaly in one or more components of the hydrological cycle. Most drought indices, however, only consider the anomaly aspect, i.e., how
unusual the condition is. In this paper, we present two drought hazard indices that reflect both the deficit and anomaly aspects. The soil moisture deficit anomaly index, SMDAI, is based on the drought severity index, DSI (Cammalleri et al., 2016, Hydrol. Process., 30, 289–301), but is computed in a more straightforward way that does not require the definition of a mapping function. We propose a new indicator of drought hazard for water supply from rivers, the streamflow deficit anomaly index, QDAI, which takes into account the surface water demand of humans and freshwater biota. Both indices are computed and
analyzed at the global scale, with a spatial resolution of roughly 50 km, for the period 1981-2010, using monthly time series of variables computed by the global water resources and the model WaterGAP2.2d. We found that the SMDAI and QDAI values are broadly similar to values of purely anomaly-based indices. However, the deficit anomaly indices provide more differentiated spatial and temporal patterns that help to distinguish the degree and nature of the actual drought hazard to vegetation health or the water supply. QDAI can be made relevant for stakeholders with different perceptions about the
importance of ecosystem protection, by adapting the approach for computing the amount of water that is required to remain in the river for the well-being of the river ecosystem. Both deficit anomaly indices are well suited for inclusion in local or global drought risk studies.

**Keywords: drought index, anomaly, soil moisture deficit, streamflow deficit, water abstraction**

## 1 Introduction

According to the Australian Bureau of Meteorology, "drought is a prolonged, abnormally dry period when the amount of available water is insufficient to meet our normal use" (BoM, 2018). This definition describes drought as both an anomaly ("less water than normal") and a deficit ("less water than required"), reflecting general non-expert notions of drought. However, most experts define drought only as an anomaly, for example, as "a lack of water compared to normal conditions

which can occur in different components of the hydrological cycle" (van Loon et al., 2016, p.3633). Assuming that humans

and other biota are accustomed to seasonal variations of water availability in the form of precipitation, soil moisture, streamflow, or groundwater storage, droughts are mostly defined by the deviation of a water quantity at a specific point in time (e.g., precipitation in May 2005) from its long-term mean or median (e.g., of all May precipitation values during the reference period 1981-2010). It is further assumed for most drought hazard indicators that humans and other biota are used to interannual variability. Therefore, drought is not defined by a percentage deviation but rather by using percentiles (e.g., precipitation in

May 2005 is less than the 10$^{th}$ percentile of all May precipitation values during the reference period) or by standardized drought indicators where the anomaly is divided by the standard deviation. **Anomaly-based drought indicators** that indicate "less water than normal" include the Standardized Precipitation Index (SPI) (Mckee et al., 1993), the Standardized Precipitation Evapotranspiration Index (SPEI) (Vicente-Serrano et al., 2010; Bergez et al., 2013), the China Z index (CZI) (Wu et al., 2001) and, for streamflow drought, the Standardized Streamflow Index (*SSFI*) (Modarres, 2007) and the percentile-based low-flow

index by Cammalleri et al. (2017).

Some researchers have quantified drought by only considering the deficit aspect of drought, i.e., by computing the difference between an optimal water quantity and the actual quantity ("less water than required"). **Deficit-based indicators** have only been derived for assessing drought risk for vegetation, as optimal water quantities can be defined by either the field capacity of the soil (Sridhar et al., 2008) or potential evapotranspiration. For the latter, the deficit is computed either as the

difference between potential evapotranspiration and precipitation (Hogg et al., 2013) or between potential and actual evapotranspiration. A drawback of these deficit-based drought hazard indicators is that they indicate strong drought in arid and (semi)arid regions, even though the vegetation in these regions is adapted to generally lower soil moisture (Cammalleri et al., 2016). Deficit-based indicators cannot be meaningfully derived for the variable precipitation only as the definition of an optimal precipitation amount depends on the user of the precipitation water. It is, however, conceptually meaningful to

determine deficits for human water supply based on the variable streamflow, defining the deficit as the difference between the demand for water from the river and the actual streamflow. To the best of our knowledge, streamflow drought has not, as yet, been characterized by a deficit-based drought indicator.

Two notable attempts in identifying and bringing together both the anomaly and deficit aspects are the Palmer Drought Severity Index (PDSI) (Palmer, 1965) and the Drought Severity Index (DSI) (Cammalleri et al., 2016). PDSI is a standardized

index developed to quantify the cumulative deficit of moisture supply in the form of precipitation as compared to demand in the form of potential evapotranspiration. Its strengths and weakness have been well investigated by Dai et al. (2004) and is extensively used in the USA to indicate meteorological droughts (Heim, 2002). DSI indicates soil moisture drought by combining the soil moisture deficit (as compared to the situation in which plant evapotranspiration is not constrained by soil moisture availability) and the anomaly of the deficit, thus indicating rare events in which plants suffer from water stress. An

anomaly-based soil moisture drought may, however, be unsuitable for indicating a drought hazard for vegetation as, in areas with high soil moisture in most years, the low interannual variability and, thus, the standard deviation, would indicate a strong

drought hazard in years with unusually low soil moisture values that are, nevertheless, still close to the optimal values and do not cause any water stress for the plants (Cammalleri et al., 2016).

Similar to the demand for soil water by plants, humans have a demand for water from rivers in situations where they
rely on river water for their water supply. About three-quarters of global water withdrawals for irrigation, cooling of thermal power plants, manufacturing and domestic use, totaling about 3700 km$^3$/a in the first decade of this century, are sourced from surface water (Döll et al., 2014). Globally, irrigation is the largest water demand sector, accounting for more than 60% of total surface water withdrawals (Müller Schmied et al., 2021; Döll et al., 2014). To date, however, streamflow drought indicators only describe the anomaly of streamflow but do not indicate whether there is enough water in the river to meet water demand.
Thus, to assess the risk of drought for human water supply from rivers, an indicator that combines the anomaly of streamflow conditions with a deficit, with respect to water demand, is desirable. In this way, the locations and times where the human water supply is at risk can be identified.

Differently from anomaly-based streamflow drought indicators, a combined analysis of streamflow anomaly and deficit requires time series information of both streamflow and water demand. This information is available from global water
resources and uses models such as WaterGAP with a spatial resolution of 0.5° (55 km by 55 km at the equator) and a monthly temporal resolution (Alcamo et al., 2003; Müller Schmied et al., 2021). Up to the present time, macro-scale drought risk assessments have included the demand for water as vulnerability indicators by using a country's average water withdrawal to water availability ratio (e.g., Meza et al., 2020).

In this study, we introduce and relate two drought hazard indicators that combine both the deficit and anomaly aspects:
one for soil moisture drought and the other for streamflow drought. In the soil moisture deficit anomaly index (SMDAI), the deficit is calculated as the difference between the soil moisture at field capacity (which allows optimal and non-water-limited plant growth) and the actual soil moisture. The SMDAI slightly modifies and simplifies the DSI introduced by Cammalleri et al. (2016). Another difference from Cammalleri et al. (2016) is that the SMDAI is computed globally, using the output of WaterGAP, rather than just for Europe. The streamflow deficit anomaly index QDAI is, to our knowledge, the first-ever
streamflow drought indicator that combines both the anomaly and deficit aspects of streamflow drought. In the case of QDAI, the deficit is computed by comparing actual streamflow to the combined human and environmental surface water demand per grid cell. QDAI focuses on determining the drought hazard for the water supply for humans, including domestic, industrial, and irrigation water demand. QDAI is constructed similarly to SMDAI and computed globally using WaterGAP. Whether QDAI should be called a drought hazard indicator, or a combined drought hazard and vulnerability indicator, is up for
discussion. However, for global-scale drought risk assessments, gridded QDAI values can be meaningfully combined with country-scale vulnerability indicators of, for example, coping capacity.

In Section 2, we describe (a) how water demand, streamflow, surface water use, and soil moisture are computed by WaterGAP 2.2d (Müller Schmied et al., 2021) and (b) the methods for calculating SMDAI and QDAI. In section 3, spatial and temporal patterns of SMDAI and QDAI are presented. In Section 4, we analyze the components of SMDAI and QDAI, compare

SMDAI to DSI, compare QDAI to a standardized streamflow indicator (*SSFI*) and discuss the limitations of the study. Finally, we draw conclusions in Section 5.

## 2. Methods and data

### 2.1 Global-scale simulation of soil moisture, soil water capacity, streamflow and human water abstraction

In this study, we use the outputs of the latest version of the global hydrological and water use model WaterGAP 2.2d
(Müller Schmied et al., 2021). WaterGAP consists of three major components: the water use models, the linking model GSWUSE and the global hydrological model (WGHM). The water use models compute water use in the five sectors: household, manufacturing, cooling of thermal power plants, livestock and irrigation. Household and manufacturing water use is computed based on national statistics (Flörke et al., 2013). The amount of water required for cooling of thermal power plants is calculated based on the location, type, and size of power plants and the annual time series of thermal electricity production
(Flörke et al., 2013). Irrigation water use is computed based on information on the irrigated area and climate for each grid cell. The irrigation model first computes cell-specific cropping patterns and growing periods and then irrigation consumptive water use, distinguishing only rice and non-rice crops (Döll and Siebert, 2002). The irrigated areas are changing over time (Siebert et al., 2015). The globally small amount of livestock water use is the only temporally constant water use and is determined from the number of livestock and livestock-specific water use values (Alcamo et al., 2003). Water use for households,
manufacturing and cooling of thermal power plants are constant throughout the year but change from year to year.

The water use models themselves do not take into account the source of the sectoral water abstractions. This is done by GWSWUSE, which computes monthly time series of 0.5° grid-cell values of human water abstractions from 1) surface water bodies (river, lakes, and man-made reservoirs) and 2) groundwater, for each of the five sectors, as well as the respective net abstractions from both sources (Döll et al., 2012). A comparison of simulated annual sectoral water abstractions per country
to independent values from the AQUASTAT database of FAO showed a rather high similarity between the two data sets (Müller Schmied et al., 2020).

Taking into account the net abstractions, i.e. the difference between water abstractions and return flows, WGHM simulates, and a daily time step, the most relevant hydrological processes occurring on the continents and computes water flows such as actual evapotranspiration, runoff, groundwater recharge and streamflow, as well as the amount of water stored
in diverse compartments such as the soil and the groundwater for all land areas, excluding Antarctica (Müller Schmied et al., 2014; Döll et al., 2003; Alcamo et al., 2003). The soil is represented as one water storage compartment that is characterized by 1) soil water capacity ($S_{max}$), which is computed as the product of land cover, specific rooting depth, and soil water capacity in the upper meter and 2) soil texture, which affects groundwater recharge (Müller Schmied et al., 2014). The temporal development of soil moisture (S) is computed from the balance of inflows (precipitation and snowmelt minus interception by
the canopy) and outflows (actual evapotranspiration and total runoff from the land). Total runoff from the land fraction of the grid cell is then partitioned into the fast surface and subsurface runoff and the diffuse groundwater recharge. Both components

are subject to so-called fractional routing to the various other storages within the 0.5° grid cell, which include the groundwater as well as lakes, wetlands, man-made reservoirs, and rivers (Döll et al., 2014). Streamflow ($Q_{ant}$) in each grid cell depends on the runoff generated within the cell, inflow from upstream grid cells as well as human water abstractions and takes into account the impact of man-made reservoirs.

WGHM is calibrated to match long-term annual observed streamflows at the outlets of 1319 drainage basins that cover ~54 % of the global drainage area, following the calibration principles provided by Müller Schmied et al. (2014), Hunger and Döll (2008), and Döll et al. (2003). In validation studies against time series of observed streamflows, WaterGAP has been repeatedly shown to be among the best-performing global hydrological models (Zaherpour et al., 2019; Zaherpour et al., 2018; Veldkamp et al., 2018). Nevertheless, there can be significant mismatches between the observed and simulated seasonality and interannual variability. "It is found that WaterGAP can simulate the low flow percentile (Q95) very well, but it can also overestimate the return period of low streamflow (Zaherpour et al., 2018).

This study uses 30-years (1981- 2010) monthly time series of WaterGAP gridded (0.5° x 0.5°) outputs for 67420 land grid cells covering all land areas of the globe except Greenland and Antarctica. These include 1) soil moisture (S) [mm], 2) streamflow ($Q_{ant}$) [km$^3$ month$^{-1}$], 3) streamflow under naturalized condition ($Q_{nat}$) [km$^3$ month$^{-1}$], assuming there are no human water abstraction or man-made reservoirs, and 4) total surface water abstractions ) [km$^3$ month$^{-1}$].In addition, the consistent dataset of soil water capacity ($S_{max}$) [mm] is utilized.

## 2.2 Computation of deficit and anomaly components of the soil moisture deficit anomaly index SMDAI

### 2.2.1 Deficit

Soil moisture deficit ($d_{soil}$) refers to the lack of water in the root zone for plants as compared to optimal growing conditions assumed to occur at soil water capacity (demand for water). $d_{soil}$ is calculated as

$$d_{soil} = \frac{S_{max} - S}{S_{max}} \tag{1}$$

where $S_{max}$ [mm] is the amount of water stored in the soil between field capacity and wilting point within the plant's root zone, S [mm] is the actual amount of soil water (soil moisture). $d_{soil}$ ranges from 0 (no deficit/stress) to 1 (extreme deficit/stress). This definition of soil moisture deficit is different from the one used in Cammalleri et al. (2016, their Eq.1) because their definition cannot be applied when using the global hydrological model WaterGAP to compute soil moisture. The deficit computation according to Cammalleri et al. (2016) requires data on soil moisture content at the wilting point and at field capacity, which is not available in WaterGAP. With our approach, which is consistent with the way of computing actual evapotranspiration from potential evapotranspiration in WaterGAP, d-values at low soil moisture saturation are lower than those of Cammalleri et al. (2016), while they are much higher at high soil moisture as Cammalleri et al. (2016) assume that deficits only occur if soil moisture is less than 50% of field capacity. Consequently, we identify very few months and grid cells

with a deficit of zero, likely less than we would do if we would have implemented the deficit definition of Cammalleri et al. (2016).

### 2.2.2 Anomaly

Assuming that vegetation is used to seasonal variations of soil moisture, the anomaly of monthly soil moisture is determined separately for each calendar month. In case of standardized drought indicators such as the SPI, a so-called z-score is computed separately for each calendar month (here using, for example, 30 monthly soil moisture deficits in the 30 January months during the period 1981-2010), by standardizing the variable using the calendar month mean and standard deviation after translating the cumulative distribution function that optimally fits the distribution of monthly values to a normal distribution (McKee et al., 1993). Thus, computation of the z-score assumes that the vegetation is adapted to both seasonal and interannual variability. Following Cammalleri et al. (2016), in this study, we express the anomaly aspect of drought not by the z-score but by deriving a so-called drought probability index (p) that can be combined with the deficit indicator to a deficit-anomaly drought hazard index.

Computation of p also starts with identifying the probability of exceedance of a certain soil moisture deficit F. Sheffield et al. (2004) found that time series of soil moisture per calendar month are best represented by the beta distribution function. The cumulative density function F of the beta distribution function can be expressed as

$$F(d_{soil}; a, b) = \frac{B(d_{soil}; a, b)}{B(a, b)} \tag{2}$$

where $a, b \geq 0$ are the shape parameters, $B(a, b)$ is the beta function and $B(d_{soil}; a, b)$ is the incomplete beta function. In this form, the b supports the range of $d_{soil} \in [0, 1]$. In this study, we could confirm the assumption made by Cammalleri et al. (2016) that the beta distribution function represents satisfactorily the distribution of $d_{soil}$, which is the same as that of the soil moisture itself. The beta cumulative distribution function was fitted to $d_{soil}$ values for each calendar month and grid cell (i.e., for each grid cell, twelve beta functions are fitted corresponding to the twelve calendar months).

Following Cammalleri et al. (2016), the next step was to derive from F a drought probability index ($p_{soil}$) that translates the probability that a certain soil water deficit status is drier than usual into the range [0, 1]. As suggested by Agnew (2000), a z-score of -0.84, which corresponds to a return period of 5 years and a $F(d_{soil})$ of 0.8, was assumed to be the threshold for drought (Table 1), for which $p_{soil} = 0$. Then, the drought probability index is calculated as

$$p_{soil} = \frac{F(d_{soil}) - 0.8}{1 - 0.8} \tag{3}$$

where $F(d_{soil})$ is the beta cumulative distribution function fitted to $d_{soil}$. If the beta cumulative distribution function is fitted to S, then $(1-F(S))$ should be used instead of $F(d_{soil})$.

Cammalleri et al. (2016) calculated $p_{soil}$ using the mode instead of median as the reference for the normal status of $d_{soil}$. The computation of $p_{soil}$ from $F(d_{soil})$ was carried out in two steps. First, for $d_{soil}$ values that are greater than or equal to the mode, a new standardized cumulative distribution function $F*(d_{soil})$ is computed (Eq. 3 in Cammalleri et al., 2016).

Subsequently, mapping of $F*(d_{soil})$ values ranging from 0.6 to 1 onto the $p_{soil}$ range of [0, 1], an exponential function (Eq. 4 in Cammalleri et al., 2016) was employed. This exponential function was developed to fit subjectively defined pairs of $F*(d_{soil})$ and $p_{soil}$ (Table 1 in Cammalleri et al., 2016). In this study, we have simplified the more complex approach of Cammalleri et al. (2016) by relying directly on $F(d_{soil})$ for mapping $F(d_{soil})$ onto $p_{soil}$ according to Eq. 3. In our opinion, there is no added value in defining an arbitrary exponential mapping function for deriving an indicator for the probability of a

drought occurrence ($p_{soil}$). Further, like most other drought researchers, we prefer the median to the mode, as among 30 deficit values, which are rational numbers, there is no true mode, i.e., no value that occurs most often. The relation between the anomaly component of SMDAI (i.e., $p_{soil}$) to the non-exceedance probability of the soil moisture deficit ($F(d_{soil})$) and the pertaining return periods, z-scores, and class names, according to Agnew (2000) as well as the anomaly component of DSI (p_DSI) are presented in Table 1. A comparison of $p_{soil}$ to p_DSI values as a function of ($F(d_{soil})$) as presented in Table 1 is

shown in Figure S1 and the slight differences between $p_{soil}$ and p_DSI, as well as DSI and SMDAI, computed with WaterGAP output for August 2003 at the global scale are presented in Figure S2. For very few grid cells, SMDAI is much larger than DSI and there are some areas where DSI is slightly larger than SMDAI. For the period 1981-2010, SMDAI is, averaged over all grid cells, 0.05 larger than DSI with according to Eq. 1.

Table 1. Relationship of the anomaly component p of SMDAI and QDAI to the non-exceedance probability of the soil moisture deficit ($F(d_{soil})$) or of streamflow ($F(Q)$), the pertaining return periods, z-scores and class names according to Agnew (2000) as well as the p-values by Cammalleri et al. (2016) to compute DSI. The class name refers to the drought conditions with z-score values that are larger than those listed in the z-score column. The equiprobability transformation technique, first suggested by Abramowitz and Stegun (1965) and utilized in Kumar et al. (2009) for calculation of the Standardized
Precipitation Index (SPI), is used to back-calculate F values from the z-score values.

| $F(d_{soil})$/ $F(Q)$ | Return period (yrs) | z-score | Drought class name | p_DSI | $p_{soil}/p_Q$ |
|---|---|---|---|---|---|
| 0.8 | 5 | -0.84 | Normal | 0 | 0 |
| 0.843 | 6.4 | -1.00 | Mild | 0.04 | 0.21 |
| 0.87 | 7.7 | -1.12 | Moderate | 0.10 | 0.35 |
| 0.9 | 10 | -1.28 | Moderate | 0.26 | 0.50 |
| 0.933 | 15 | -1.50 | Moderate | 0.54 | 0.68 |
| 0.95 | 20 | -1.64 | Severe | 0.72 | 0.75 |
| 0.97 | 33.3 | -1.88 | Severe | 0.89 | 0.85 |
| 0.9775 | 40 | -2.00 | Severe | 0.93 | 0.88 |
| 0.99 | 99 | -2.33 | Extreme | 0.99 | 0.95 |
| 0.995 | 200 | -2.57 | Extreme | 0.997 | 0.97 |
| 0.998 | 500 | -2.88 | Extreme | 0.999 | 0.99 |
| 1 | -- | ~ -4.00 | Extreme | ~ 1 | ~ 1 |

**2.3 Computation of deficit and anomaly components of the streamflow deficit anomaly index QDAI**

**2.3.1 Deficit**

Similar to the soil moisture deficit, the streamflow deficit ($d_Q$) is calculated as the demand for water minus the supply divided by demand. It refers to the amount of streamflow that is lacking to satisfy the surface water demand of both humans

and the river ecosystem. $d_Q$ is computed as

$$d_Q = \frac{(WU_{sw} + EFR) - Q_{ant}}{WU_{sw} + EFR} \tag{4}$$

where $WU_{sw}$ [km³ month⁻¹] is water abstraction from surface water bodies, derived as the sum of water abstractions for irrigation, livestock, cooling of thermal power plants, manufacturing and household use. $Q_{ant}$ [km³ month⁻¹] is the streamflow and EFR [km³ month⁻¹] is the environmental flow requirement, i.e., the surface water demand of the river ecosystem. Following Richter et al. (2012), EFR is calculated for each calendar month as 80% of the mean monthly streamflow under the

naturalized condition ($\overline{Q_{nat}}$), assuming that 80% of the natural mean monthly streamflow that would have occurred in the river without human water use and man-made reservoirs needs to remain in the river for the well-being of the river ecosystem.

. Differing from $S_{max}$, which represents the vegetation demand for soil water, the streamflow demand is temporally variable. $d_Q$ is, like $d_{soil}$, in the range of 0 (no deficit/stress) to 1 (extreme deficit/stress); if $d_Q$ is less than 0 or $WU_{sw}$ equals 0, then $d_Q$ is set to 0. To explore how assumptions about EFR and, thus, total surface water demand affects QDAI, we set EFR
to be alternatively equal to half of $Q_{nat}$, or zero (Section 3.2 and Section 4.2). These alternatives represent situations in which humans wish to protect freshwater biota less, or not at all, so the total surface water demands and consequently streamflow deficits are lower.

### 2.3.2 Anomaly

Streamflow anomaly ($p_Q$) is computed based on the interannual variability of monthly aggregated streamflow ($Q_{ant}$)
values for each calendar month. We select to consider the anomaly of streamflow ($Q_{ant}$) instead of the anomaly of the streamflow deficit ($d_Q$) as the temporal variability including long-term trends of the water demand prevented us, for most grid cells with relevant water demand, from identifying a standard distribution function for the time series of $d_Q$. Furthermore, the methodological consistency between the calculation of $p_Q$ and $p_{soil}$ is maintained, as the anomaly of soil moisture deficit ($d_{soil}$) is equal to the anomaly of soil moisture (S) [mm].

In some regional streamflow drought studies (Langat et al., 2019; Sharma and Panu, 2015; Lorenzo-Lacruz et al., 2010; López-Moreno et al., 2009), the standard cumulative distribution function Pearson type III was used to fit monthly streamflow values. However, Svensson et al. (2017) rightly pointed out that the Pearson type III distribution function with a lower bound at zero is reduced to the gamma distribution function. The cumulative density function F of the gamma distribution function can be expressed as

$$F(Q_{ant}; a, b) = \frac{g(Q_{ant}; a, b)}{G(a)} \tag{5}$$

where $a, b \geq 0$ are the shape parameters, $G(a)$ is the gamma function and $g(Q_{ant}; a, b)$ is the incomplete gamma function; in this form the gamma distribution supports $d > 0$. Taking into account that streamflow drought occurs when a certain streamflow value is not exceeded, while in the case of $p_{soil}$ a soil moisture drought occurs when a certain soil moisture deficit is exceeded, the drought probability index for streamflow drought $p_Q$ is computed as

$$p_Q = \frac{\left(1 - F(Q_{ant})\right) - 0.8}{1 - 0.8} \tag{6}$$

### 2.4 Combining deficit and anomaly to compute SMDAI and QDAI

Water deficits ($d_{soil}$ and $d_Q$) and anomalies ($p_{soil}$ and $p_Q$) are combined into single deficit anomaly indicators (SMDAI and QDAI) based on the desired indicator characteristics as elaborated by Cammalleri et al. (2016). The combined

drought indicator should be zero if there is either no deficit- or no anomaly-based drought. It should be equal to p and d if p and d are the same, while it should have lower values if either d or p is close to zero. Thus, following Cammalleri et al. (2016)

$$\text{SMDAI} = \sqrt{p_{soil} \cdot d_{soil}} \tag{7}$$

and accordingly

$$\text{QDAI} = \sqrt{p_Q \cdot d_Q} \tag{8}$$

Both SMDAI and QDAI values range from 0 to 1, where 0 corresponds to no drought hazard and 1 corresponds to extreme drought hazard. The indicator values are put into classes and coinciding drought classifications according to Table 2.

Table 2. SMDAI and QDAI range corresponding to drought classes.

| SMDAI range /QDAI range | Drought conditions |
|---|---|
| $0 < \text{SMDAI} < 0.25$ | Mild |
| $0.25 \geq \text{SMDAI} < 0.5$ | Moderate |
| $0.5 \geq \text{SMDAI} < 0.75$ | Severe |
| $\text{SMDAI} \geq 0.75$ | Extreme |

**2.5 Fitting standard cumulative functions**

Out of the total 67420 WaterGAP land grid cells, only 57043 grid cells were considered in this study. Grid cells with barren or sparsely vegetated land cover, based on the MODIS-derived static land cover input map used in WGHM (Müller Schmied et al., 2014), together with grid cells in Greenland, were not considered. For each of these grid cells and each calendar month, we determined the best fitting beta and gamma cumulative distribution functions for monthly $d_{soil}$ and $Q_{ant}$, respectively, by utilizing a combination of functions from the R packages gamlss, gamlss.dist, extremeStat and fitdistrplus.

However, as tested by the one-sample Kolmogorov–Smirnov test (KS-test) at the 0.05 significance level, for 27.12% of the grid cells in the case of $d_{soil}$ and 39.94% in the case of $Q_{ant}$, the fits were rejected for all 12 calendar months. An example of an accepted grid cell and a rejected grid cell of the beta distribution function are shown in Figure S3. In the rejected grid cells, the probability of non-exceedance F is determined directly from the time series of 30 monthly values using the R function empirical cumulative distribution function (ECDF). The ECDF is a step function that increases by 1/30 at each of the 30 $d_{soil}$

values of SMDAI or $Q_{ant}$ values of QDAI (Figure S3 left). The computed F value of a specific $d_{soil}$ or $Q_{ant}$ value is the fraction of all 30 $d_{soil}$ or $Q_{ant}$ values that are less than, or equal to, the specific $d_{soil}$ or $Q_{ant}$ value. Figure S4 shows the grid cells where ECDFs had to be used to compute F.

**2.6 Standardized Streamflow Index**

We compared QDAI with well-established anomaly-based drought indicator Standardized Streamflow Index (*SSFI*) introduced by Modarres (2007). SSFI is computed separately for each calendar month, similar to the Standardized Precipitation Index (SPI) (Mckee et al., 1993), as

$$SSFI = \frac{Q_{anti} - \overline{Q_{ant}}}{\sigma} \tag{9}$$

where $Q_{anti}$ [km$^3$ month$^{-1}$] is the streamflow value at time interval $i$, $\overline{Q_{ant}}$ is the long-term mean of the streamflow values and

$\sigma$ is the standard deviation of the streamflow values.

**3 Results and discussions**

**3.1 SMDAI**

        The relations between $d_{soil}$, mean monthly ($d_{soil\_mean}$), $p_{soil}$ and SMDAI are further clarified by the time series of these variables in Figure 1 for two grid cells with rather different characteristics: a grid cell in Germany (42.25N, -121.75 E,

left panels in Figure 1) and one in northeast India (88.25 E,27.25 N, right panels in Figure 1).The values of $d_{soil}$ in the German grid cell show, on average over the whole reference period, high deficits in the summer months and low deficits only in 1-2 winter months (dashed grey line). According to the definition of $p_{soil}$, an anomaly-based drought hazard, as indicated by $p_{soil} > 0$ (blue line), occurs only if the actual soil moisture deficit (green line) is much higher than the mean calendar month values $d_{soil\_mean}$; per definition, this is the case in only 1 out of 5 years (Eq. 3 and Table 1). According to Eq. 7, SMDAI is always

between $p_{soil}$ and $d_{soil}$. In the German cell, an anomaly-based drought occurred during the unusually dry, but still low deficit, winter months of 2006, resulting in an SMDAI value that was much smaller than $p_{soil}$. During the Central European (CEU) summer drought of 2003, SMDAI was approximately equal to $p_{soil}$. Thus, SMDAI appropriately indicates that anomalously low soil moisture during generally wet winter months is less of a hazard to vegetation than the same anomaly would be during generally dry summer months. The grid cell in northeast India is characterized by a low seasonality of soil moisture and a

generally very high soil water content. Even for some unusually dry months (with high $p_{soil}$), $d_{soil}$ remains almost always below 0.25. Due to the low deficit, even in cases of high $p_{soil}$, SMDAI is much smaller than $p_{soil}$ during all drought events indicated by $p_{soil}$. When comparing temporally averaged drought hazards between the two grid cells, SMDAI would indicate a relatively higher drought hazard for the German grid cell than for the Indian grid cell, which would not be the case if a purely anomaly-based indicator, such as $p_{soil}$, were used as the drought hazard indicator.

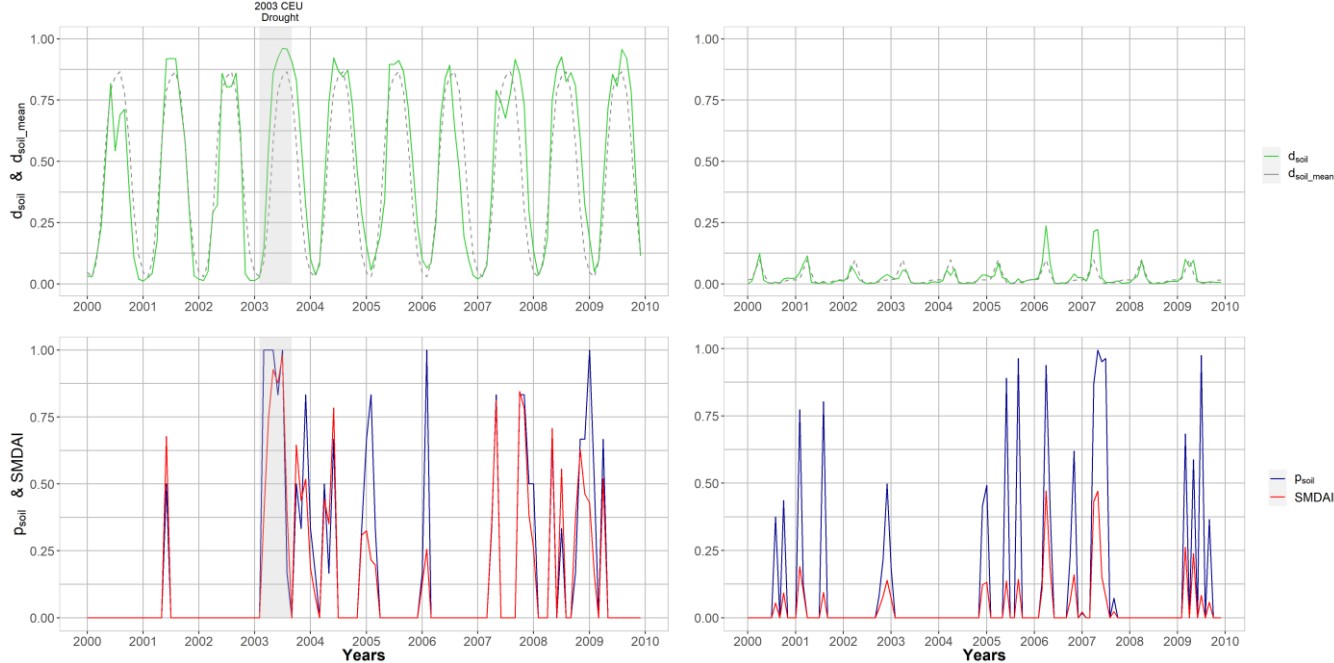

Figure 1. Soil moisture drought hazard: example of a time series (2000 – 2010) of monthly $d_{soil}$ and mean seasonality of soil moisture deficit, $p_{soil}$ and SMDAI for a cell in Germany (left) and northeast India (right). The central European (CEU) drought in 2003 is indicated.

The relationship between SMDAI, $p_{soil}$, and $d_{soil}$ can be further explored by using global indicator maps for a specific month, e.g., August 2003 (Figure 2). WaterGAP computes soil moisture deficits of 75% or more in most grid cells, while low deficits occur only in a few areas, where August belongs to the rainy season, e.g., the Sahel region and the monsoon areas in India (Figure 2a). In each grid cell, $p_{soil}$ is, per definition, zero in 80% of all August months. Therefore, in any month, approximately 80% of the grid cells indicate no drought and $p_{soil}$ equals 0 (Figure 2b). Only grid cells with a non-zero $p_{soil}$ have a non-zero SMDAI (Figure 2c). For example, southeast India shows extremely high $d_{soil}$ values, but as there is no anomalously high soil moisture deficit except for a few grid cells where $p_{soil}$ is mostly zero, SMDAI is also mostly zero. Thus, no soil moisture drought hazard is indicated. The difference between SMDAI and $p_{soil}$ is shown in Figure 2d. In most grid cells with differences, SMDAI is higher than $p_{soil}$ due to high $d_{soil}$. Focusing on central Europe, SMDAI (in Figure 2c) correctly indicates the summer drought of 2003, documented in the EM-DAT International Disaster Database (http://www.emdat.be), the European Drought Reference database (http://www.geo.uio.no/edc/droughtdb) and in Spinoni et al. (2019). The location of grid cells from Figure 1 is represented in Figure 2a with blue points drawn at the center of each grid cell. During northern hemisphere winter months, soil moisture deficits are lower, for example, in Europe and the eastern part of North America, but high in most snow-dominated northern high-latitude regions (as no liquid water enters the soil), with corresponding effects for the relationship between $p_{soil}$ and SMDAI (see Figure S5 showing the drought situation in December 1999). In Europe and the eastern part of North America, for example, SMDAI is smaller than $p_{soil}$ (Figure S5d).

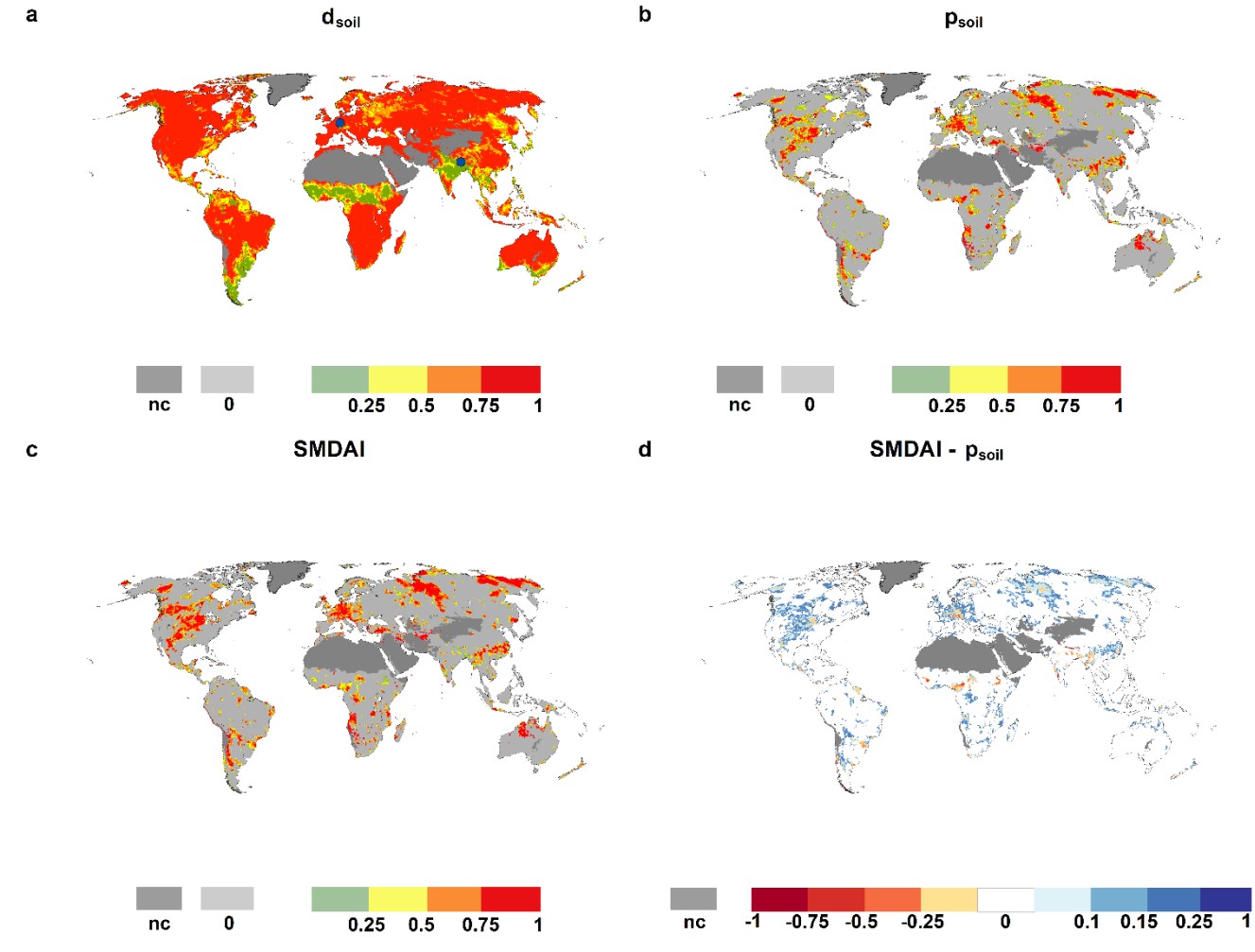

Figure 2. Global maps of $d_{soil}$, $p_{soil}$, SMDAI and the difference between SMDAI and $p_{soil}$ for August 2003. Blue points in (a) represent the location of German and Indian grid cells from Figure 1 and nc are grid cells that are not computed due to land cover.

Figure 3 shows the frequency of occurrence of the four SMDAI drought classes specified in Table 2 and of the no-drought condition (SMDAI = 0) during the reference period 1981-2010. SMDAI is zero in about 80% of the cases, following $p_{soil}$ as monthly soil moisture almost never reaches the maximum soil moisture capacity. Extreme soil moisture drought hazards occur with a relatively high frequency in the northwestern parts of Australia and southeastern parts of Africa. Regions with mostly low soil moisture deficits, such as central and eastern European countries and the eastern USA, show very low occurrence frequencies of extreme drought hazards and more often than other regions a moderate drought hazard (Figure 3b). Snow-dominated regions, such as parts of Russia and Canada, show a relatively high frequency of extreme soil moisture

droughts due to the high values of simulated soil moisture deficits created by the lack of liquid water to infiltrate the soil during the winter months and the temperature-driven seasonal shifts of snow melts and, thus, infiltration of water into the soil.

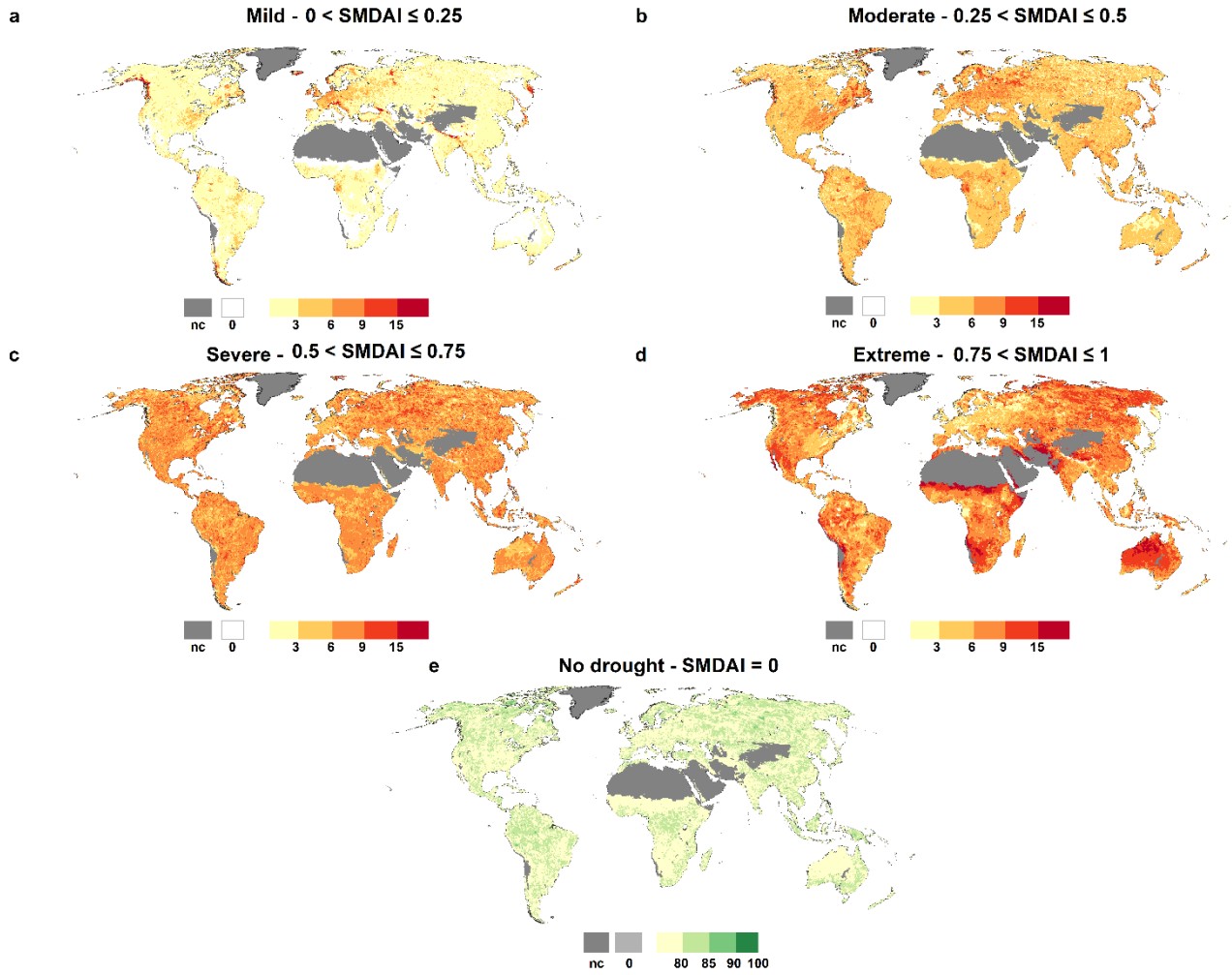

Figure 3. Frequency of occurrence [%] of different soil moisture drought classes during the period 1981-2010, as defined by SMDAI (Table 2) and nc are grid cells which not computed due to land cover.

## 3.2 QDAI

QDAI indicates the drought hazard for surface water supply required for satisfying human water demand ($WU_{sw}$), assuming the water suppliers also take into consideration the water demand by freshwater biota (EFR). The deficit component of QDAI ($d_Q$) is the relative difference between the total surface water demand and streamflow, while the anomaly component ($p_Q$) is based on the unusualness of streamflow. QDAI depends on more individual variables (i.e., $WU_{sw}$, $Q_{ant}$ and EFR) than

SMDAI (i.e., S and $S_{max}$). Figure 4 shows their relation for two grid cells with different characteristics of human surface water demand as compared to streamflow. In the grid cell in the western USA, where streamflow of the Klamath River is observed in Keno (42.25N, -121.75 E, left panels of Figure 4), water demand (mostly for irrigation, with a mean of 0.038 km$^3$ month$^{-1}$) is high compared to the relatively small streamflow (0.105 km$^3$ month$^{-1}$). In the grid cell in Germany, human surface water demand of 0.056 km$^3$ month$^{-1}$ is small as compared to the rather high streamflow of 4.6 km$^3$ month$^{-1}$ of the Rhine at Mainz (49.75 N, 8.25 E, right panels of Figure 4).

In the US grid cell, the difference between the mean monthly streamflow under the naturalized condition $\left(Q_{nat\_mean}\right)$ and mean monthly simulated streamflow ($Q_{ant\_mean}$) is high, especially in the growing period, due to large anthropogenic abstractions of streamflow water in the drainage basin of the grid cell (observed in the topmost plot). While the observed ($Q_{ant\_obs}$) and simulated ($Q_{ant}$) streamflow shows a reasonable correlation, WaterGAP appears to overestimate streamflow depletion by human water use in the summers. Characterized by a high seasonality, anthropogenic surface water demand, $WU_{sw}$ (dashed grey line in center plot) and total surface water demand (i.e., $WU_{sw}$ + EFR_0.8, orange line in center plot) result in very high deficits $d_Q$ (green line of the bottom plot) during almost every summer. However, there are only a few months with drought as identified by the anomaly-based drought hazard $p_Q$ exceeding zero (dark blue line). This occurs because the decade shown in Figure 4 happens to be a very wet decade compared to the whole reference period. Another reason is that more than 20% of the years show zero streamflow in the calendar months August and September such that $p_Q$ is zero in all 30 August and September months of the reference period, i.e. no drought is indicated even in case of zero streamflow (see left panel of Figure S7). Due to the large deficit values, $p_Q$ is almost always smaller than $d_Q$ in this US grid cell.

In the German grid cell (right panels in Figure 4), the relatively low anthropogenic surface water abstractions result in almost identical values of $Q_{nat\_mean}$ and $Q_{ant\_mean}$ (lines overlap in the top plot), and total surface water demand is very similar to EFR (lines overlap in the center plot). Non-zero $d_Q$ values (bottom plot) are mainly computed if $Q_{ant}$ is lower than EFR, such as during the central European drought of 2003. It is reasonable to consider this type of situation as a drought hazard as water supply companies would have to stop any surface water abstraction if they wished to protect the river ecosystem. Different from the US grid cell, droughts are rather equally distributed over all decades of the reference period in the German grid cell but the summers of 2003 and 2005 suffer from the most severe droughts of the reference period, in line with expected dryer summer due of climate change. Even if taking into account EFR as 80% of $Q_{nat\_mean}$ (EFR$_{0.8}$), the total surface water demand is so low that in contrast to the US cell, $d_Q$ is always smaller than $p_Q$.

Assumptions about the magnitude of EFR have a strong impact on $d_Q$ and thus QDAI of all grid cells except those with very high surface water abstractions such as the US cell. If the water demand of the ecosystem were assumed to be only 20% of $Q_{nat\_mean}$ (EFR_0.2 ) instead of 80% of $Q_{nat\_mean}$, $d_Q$ decreases somewhat in the US cell but reduces to zero during the whole reference period in the German cell (Figure S6). Therefore, water suppliers in the German grid cell would not suffer from any drought hazard (as indicated by QDAI) and would not have to decrease their surface water abstractions even during a drought similar to the 2003 central European drought.

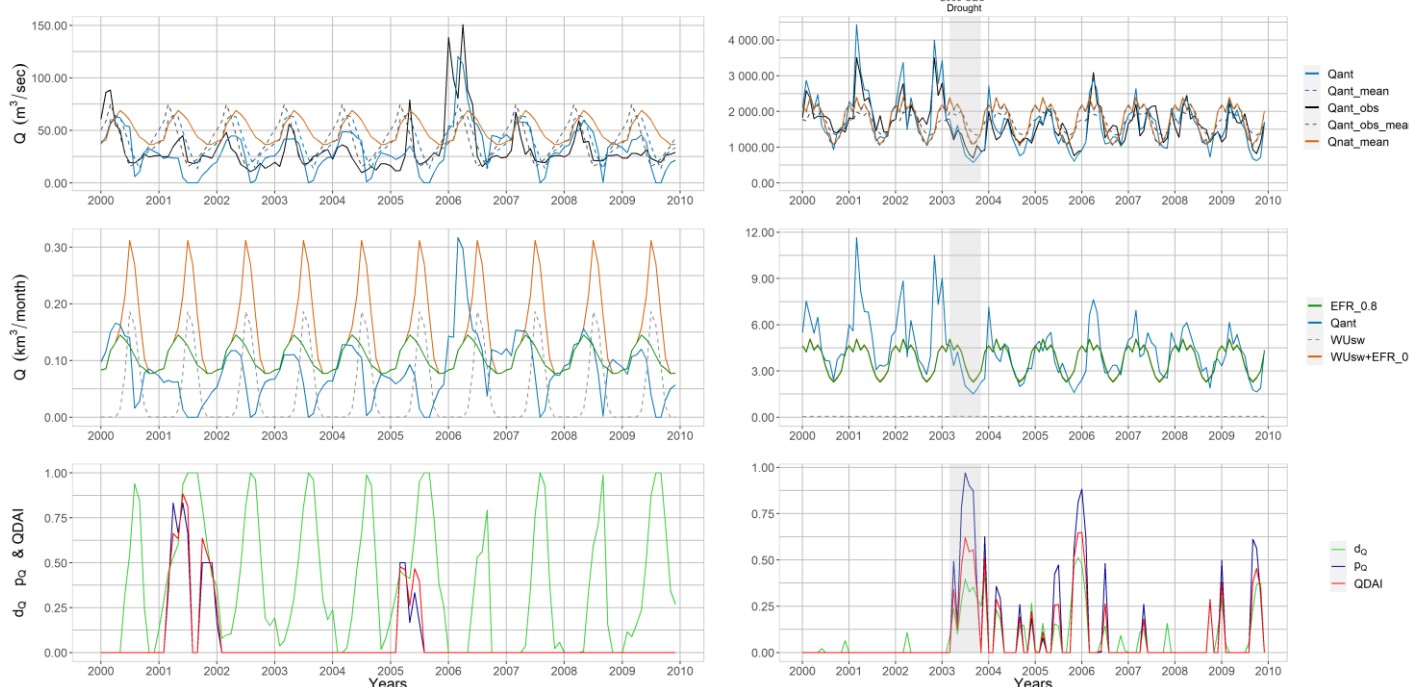

Figure 4. Streamflow drought hazard: example of a time series (2000 – 2010) of monthly surface water demand, surface water supply, and mean seasonality of surface water supply, as well as $d_Q$, $p_Q$ and QDAI (bottom) for a cell in the USA (left) and Germany (right).

The global streamflow drought hazard maps for August 2003 (Figure 5) help to illustrate the global variations of QDAI as a function of its components $p_Q$ and $d_Q$, which again depends on the human surface water demand $WU_{sw}$. Streamflow deficits are not restricted to areas with high mean annual $WU_{sw}$ during the period 1981-2010 (Figure 5a), but can be greater than 75% in regions such as South Africa were $Q_{ant}$ is low (Figure 5b). Different from soil moisture drought, $p_Q$ and $d_Q$ are strongly correlated (Figure 5c). This is due to the fact that total surface water demand is dominated in many grid cells by EFR, which is a fraction of $Q_{nat}$. In the EFR-dominated cells, the mean monthly $Q_{ant}$ is very similar to the mean monthly $Q_{nat}$, such that $d_Q$ is then approximately the difference between mean monthly $Q_{ant}$ and $Q_{ant}$; this difference is also the basis for computing by $p_Q$ (Figure 5d). QDAI is mostly less than $p_Q$ (Figure 5e). The 2003 central European drought hazard for the surface water supply for humans (Figure 5d) is, at least in many parts of Germany, less pronounced than the soil moisture drought hazard for vegetation (Figure 2c). Figures 5c-e also indicate the grid cells with $Q_{ant} = 0$. If streamflow in a grid cell is zero in 20% or more of all August months (left panel of Figure S7), $p_Q$ and thus QDAI is zero because the zero streamflow is not an anomaly that occurs in less than 1 out of 5 years.

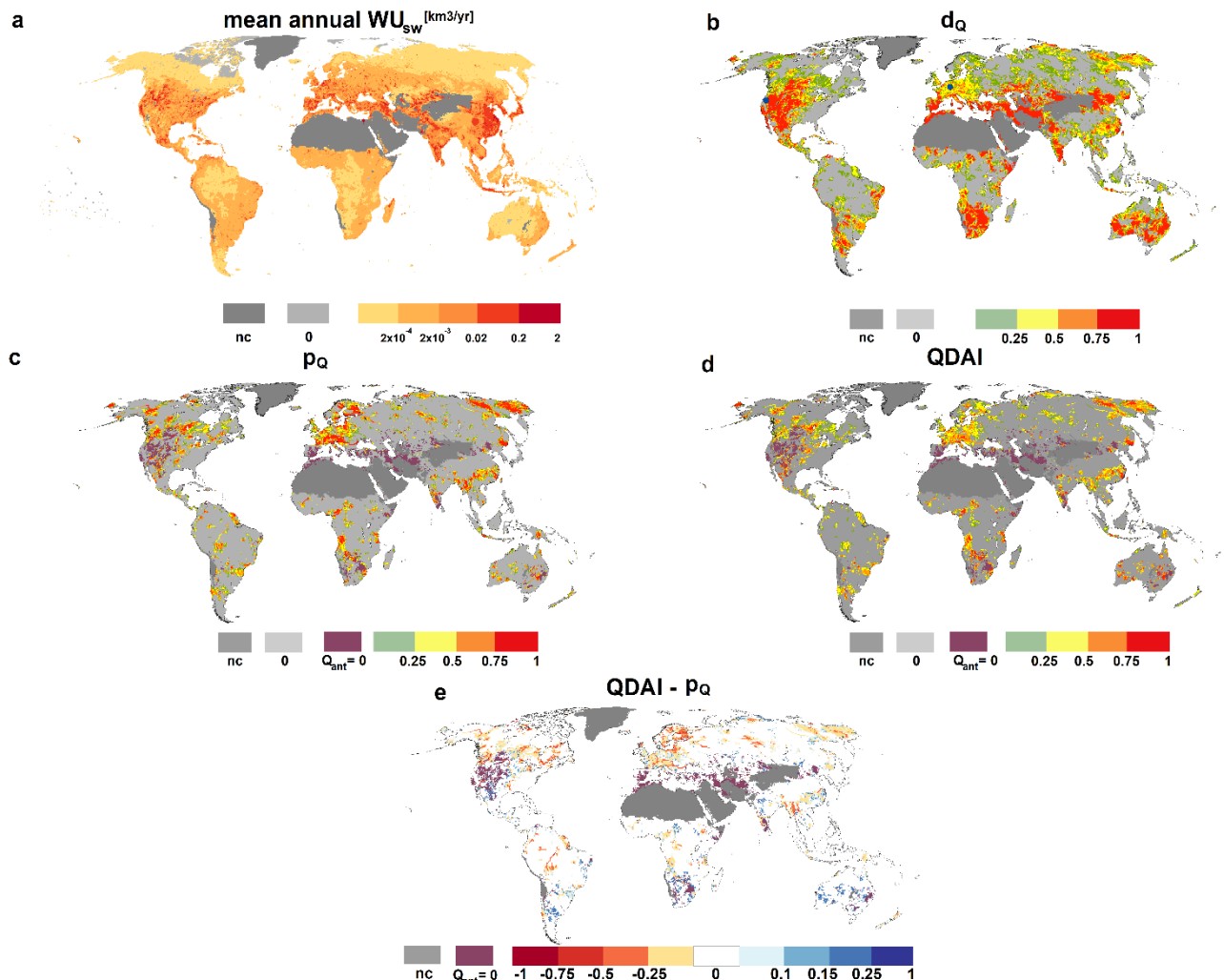

Figure 5. Global maps of mean annual $WU_{sw}$, $d_Q$, $p_Q$, QDAI and the difference between QDAI and $p_Q$ for August 2003. Blue points in (b) represent the location of the German and USA grid cells from Figure 4. Grid cells with $Q_{ant} = 0$ are indicated; nc: QDAI is not computed due to land cover.

In contrast to SMDAI, the frequency of occurrence of no-drought conditions according to QDAI (Figure 6) is larger than 80% in grid cells particularly with large rivers and barely any human water use, such as the Amazon River in South America, the Congo River in Africa, and the Ob River in Russia (Figure. 6e), where the deficit is often zero. Besides, grid cells with intermittent flows also show a high percentage of no-drought conditions, as for any calendar month with at least six months without streamflow $p_Q$ is always equal to zero (Figure S7). In these grid cells, no-drought conditions occur in the case of zero streamflow. This type of intermittent grid cells, where $Q_{ant} = 0$ for at least 20% of the months of any calendar month are marked separately in Figures 6c-e. Extreme streamflow drought hazard for human water supply (Figure 6d) occurs most often in regions with high streamflow deficits (compare Figure 5b), such as South Africa and parts of southeastern

Australia, i.e., regions with low streamflow and relatively high surface water abstractions, mainly for irrigation (Figure 5a). Regions with low water human surface water abstractions such as northern Canada and the Amazon and Congo basins show an exceptionally high occurrence of mild drought hazards (Figure 6a).

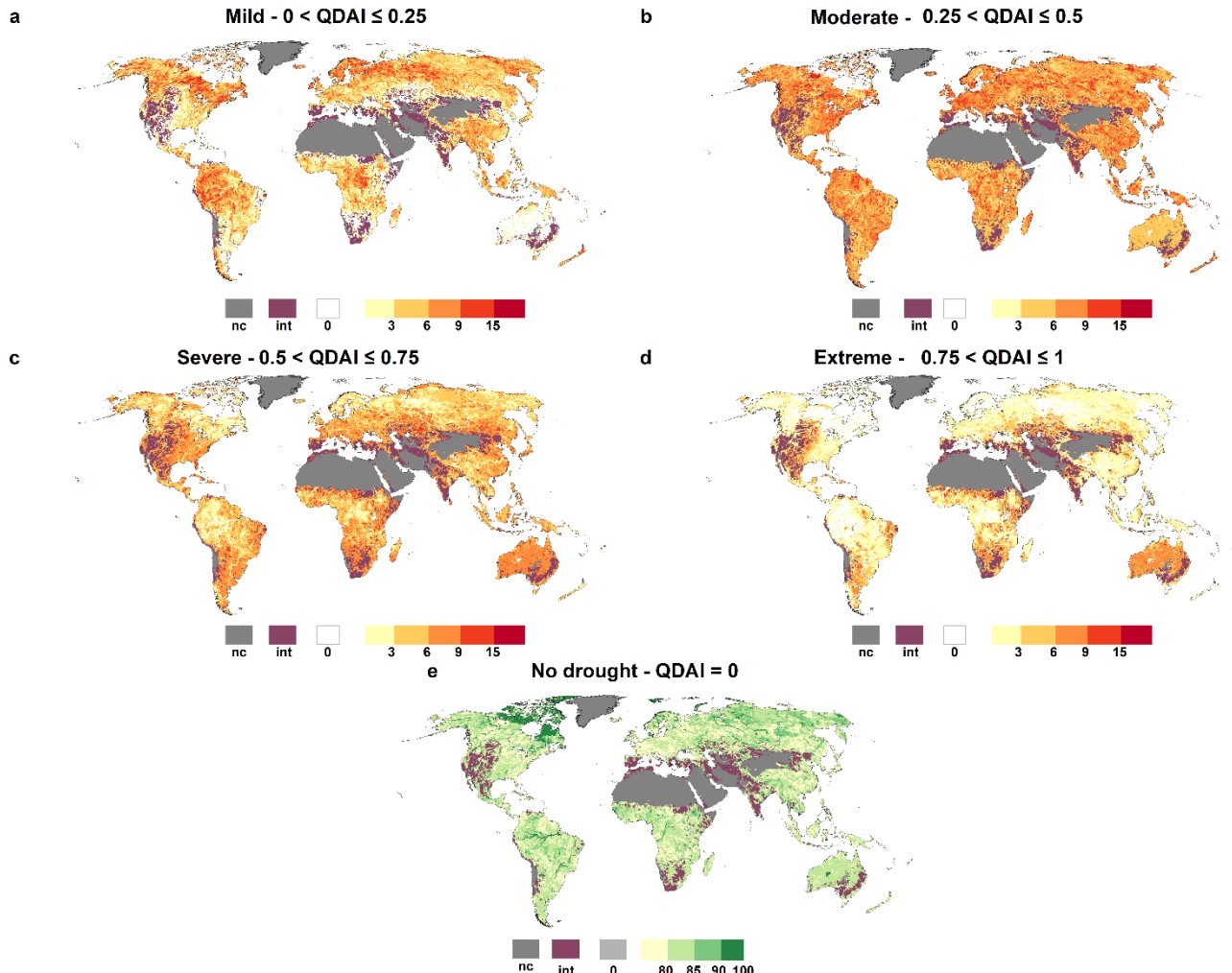

Figure 6. Frequency of occurrence [%] of different streamflow drought classes during the period 1981-2010 as defined by QDAI (Table 2). Grid cells where for any calendar month there are at least six months with $Q_{ant}$ = 0 are indicated as int and grid cells which are not computed due to land cover are indicated as nc.

### 3.3 Sensitivity of SMDAI to the Smax values assumed in WaterGAP

$S_{max}$ is one of the key components for computing SMDAI. WaterGAP calibration and validation studies have indicated that $S_{max}$ maybe underestimated in WaterGAP by a factor of two or more (Hosseini-Moghari et al., 2020). In order to understand the sensitivity of SMDAI to changes in $S_{max}$, we ran a version of WaterGAP in which $S_{max}$ was doubled ($S_{max2}$).

Figure 7 presents global maps of $d_{soil\_Smax2}$ (Figure 7a), $p_{soil\_Smax2}$ (Figure 7c), and SMDAI$_{smax2}$ (Figure 7e) for August 2003, and the change in each parameter with respect to the standard WaterGAP output i.e., the difference between parameter computed using $S_{max2}$ and $S_{max}$ (Figure 7b, 7d, and 7f). With doubled $S_{max}$, mean monthly soil moisture increases, too. In most grid cells, the soil moisture deficit increases as compared to standard $S_{max}$ (Figure 7b). Differences are mostly small except for scattered grid cells in which the soil moisture deficit decreases by more than 50 percentage points. Such cells are also found in central Europe where, under the heavy drought conditions of August 2003, computed deficits $d_Q$ are generally smaller in the case of doubled $S_{max}$; in this region, $p_{soil}$ increases in the case of doubled $S_{max}$ (Figure 7d). Globally, $p_{soil}$ increases or decreases in some grid cells by more than 50 percentage points. Equally, for SMDAI, the sensitivity to doubled $S_{max}$ is low for most grid cells but can be greater for a few (Figure 7e).

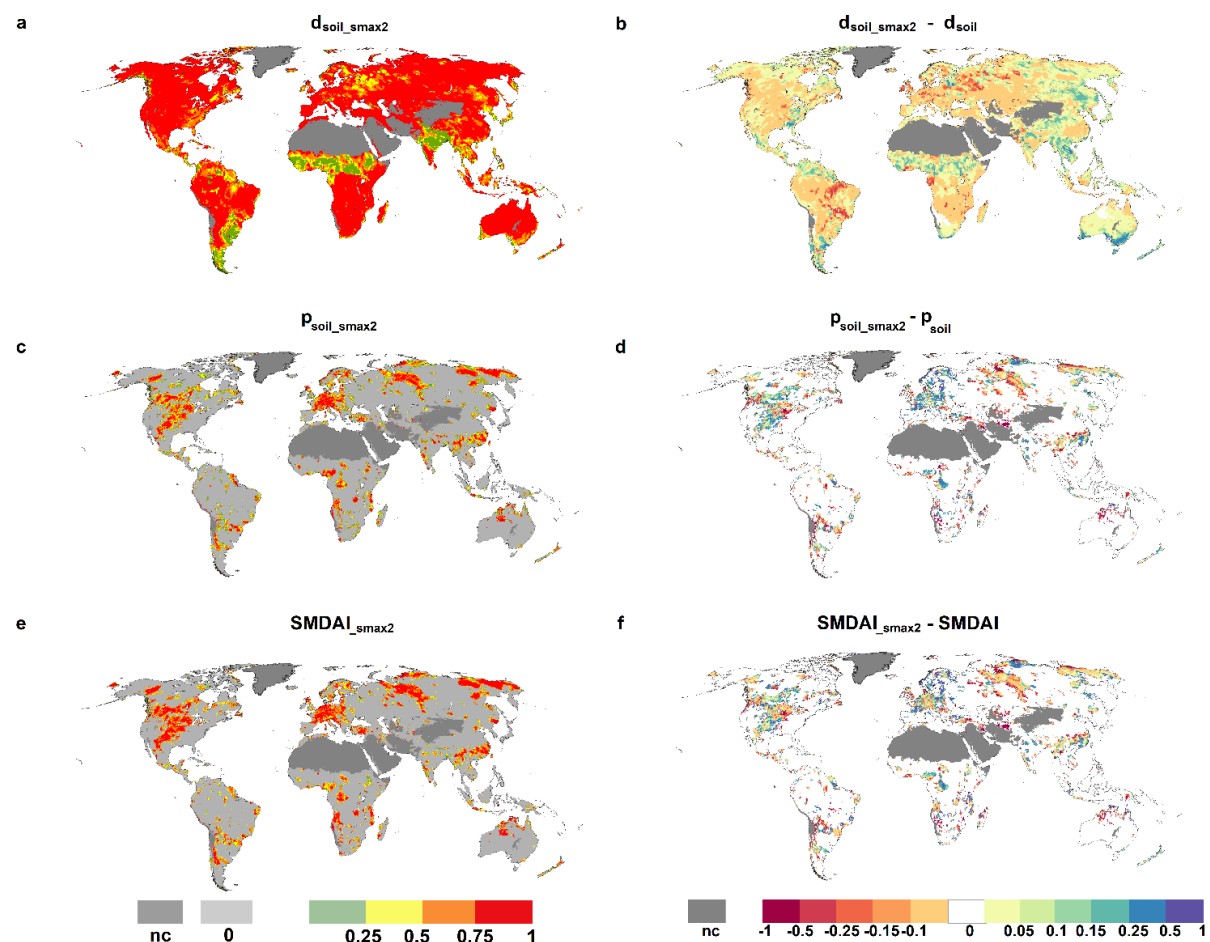

Figure 7. Spatial representation of $d_{soil}$, $p_{soil}$ and SMDAI computed with $S_{max2}$ are presented in the left panel, and, in the right panels, are the differences in these $d_{soil}$, $p_{soil}$, and SMDAI compared to the results computed with the standard version of WaterGAP for August 2003 as well as nc are grid cells which not computed due to land cover.

**3.4 Sensitivity of QDAI to different assumptions about EFR**

The streamflow drought hazard for water supply indicated by QDAI depends on how EFR is defined. In Figure 8, we compare the global distribution of QDAI values among the 57043 0.5° grid cells, assuming that either 80% or 50% of mean monthly natural streamflow is required to remain in the river for the well-being of the river ecosystem, or that there is no EFR at all that needs to be considered when the decisions about river water abstractions are made. We distinguish between humid and (semi)arid grid cells (Figure S8) and consider the two months of August and December 2003 as well as all 360 months of

the reference period. The QDAI distributions are very similar for all three time periods. The boxplots show that a drought hazard in humid areas is only identified if the existence of an EFR is acknowledged. If water suppliers in humid areas assume that all water in the river can be abstracted, they will very rarely be unable to satisfy their demand. In humid grid cells, QDAI increases strongly with the selected EFR, which means that with increasing consideration of the water requirements of the river ecosystems, drought hazards to the water supply increase, i.e., there are more situations where water abstractions would have

to be reduced to keep enough water in the river for the ecosystems to thrive. In (semi)arid regions, QDAI is already very high, even without acknowledging any water requirement of the river ecosystem. This is due to an often high surface water demand as compared to naturalized streamflow, in particular as crop production requires irrigation. Like in humid regions, QDAI increases with increasing EFR. The slightly higher median QDAI values in August 2003 than in December 2003 reflect the larger amount of humid grid cells in the northern hemisphere. Figure 8 shows that water suppliers in (semi)arid and arid regions

suffer much more strongly from drought hazards than water suppliers in humid areas due to the much higher ratio of water demand to streamflow.

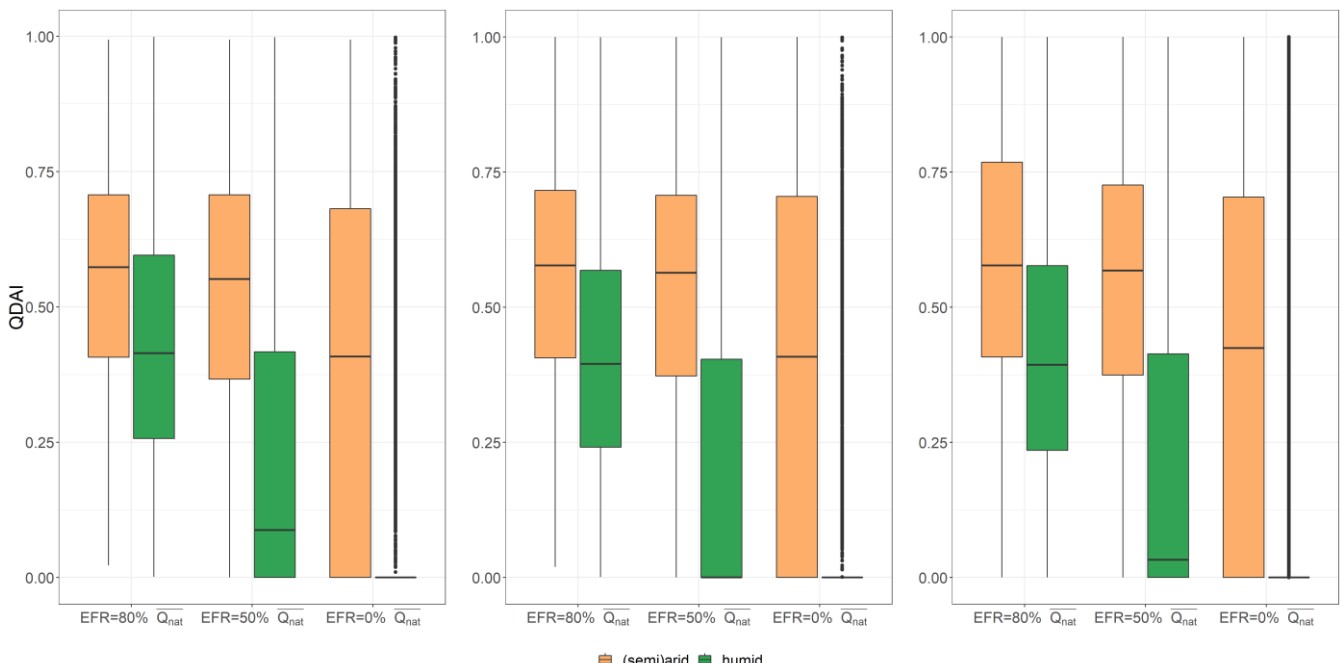

Figure 8. Global distribution of QDAI in August 2003 (left), December 2003 (middle) and for all 360 months of the reference
period (right), computed with alternative assumptions about **EFR** for grid cells with humid and (semi)arid conditions. Grid
cells where all three **EFR** assumptions result in QDAI = 0 are not included.

Further differences between QDAI values computed for alternative EFR are explored for two widely known drought

events, the South Asian drought of 2009 (Neena et al., 2011) and the North American drought of 2002 (Seager, 2007). Figure

9 presents the spatial extent of both the droughts detected by QDAI at a continental scale (left panels of figure 9) for August

2009 and March 2002, respectively. Time series plots (right panels of Figure 9) for an Indian grid cell (75.75 E, 24.75 N top

panel), as well as another for a USA grid cell (-110.75 E, 44.25 N bottom panel), provide a better understanding of the

sensitivity of QDAI to EFR. As expected, QDAI values calculated with EFR = 0 (green) are lower and drought periods shorter

than if it is assumed that water needs to remain in the river for the well-being of the ecosystems. Interestingly, short but severe

drought in the Indian grid cell in 2002, 2006, and 2010 have almost equal QDAI values for all three EFR alternatives.

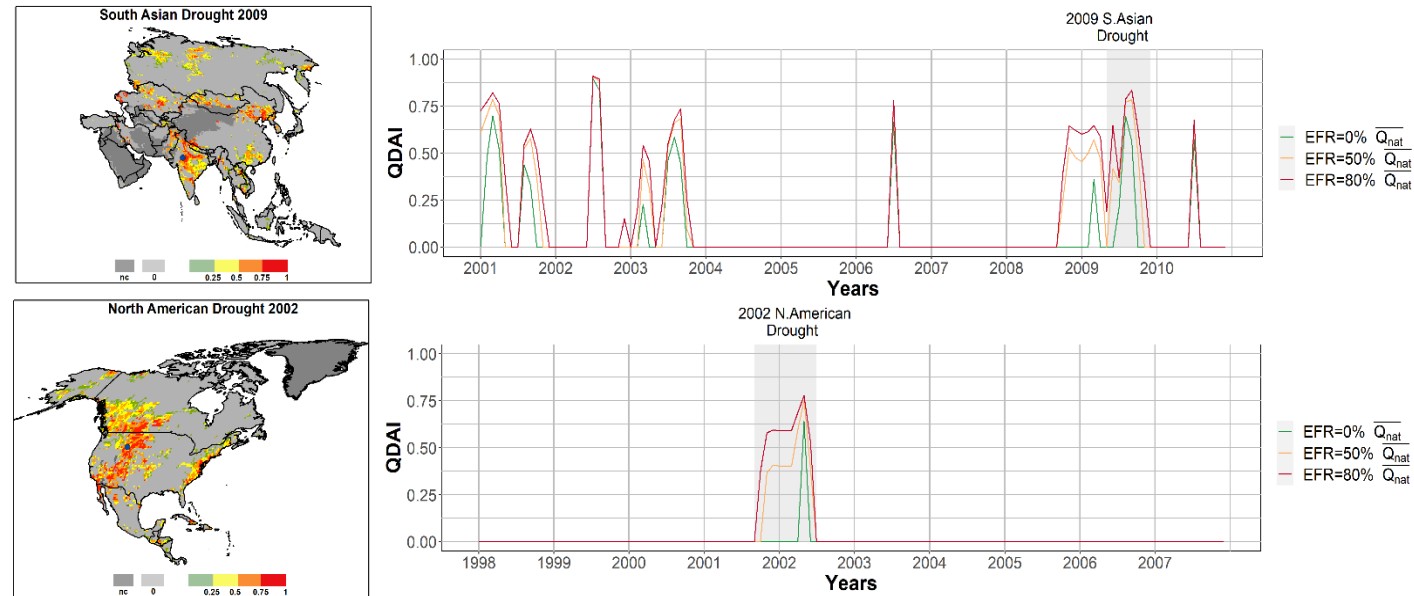

Figure 9. Continental maps of QDAI for Asia and Northern America for August 2009 and March 2002 respectively (left panels) with blue points showing the location of the Indian and USA grid cells. Time series of different QDAI with alternative **EFR** (right panels) for Indian grid cell for 2001-2010 and USA grid cell for 1998 – 2007 and nc are grid cells which are not computed due to land cover.

### 3.5 Comparing QDAI to the Standardized Streamflow Index (SSFI)

Like $p_Q$, *SSFI* (see section 2.6) assumes biota and humans are accustomed to the seasonal and interannual variability of the streamflow. In order to quantify the added value of QDAI, we compared QDAI values to *SSFI* values computed with a 1-month timescale. The anomaly of streamflow in *SSFI* was computed in the same manner as for $p_Q$, by fitting the gamma cumulative distribution function for monthly $Q_{ant}$. It was then transformed into Gaussian distribution by calculating the mean, standard deviation, as well as using the approximate conversion provided by Abramowitz and Stegun (1965); this is also used by Kumar et al. (2009). Figure 10 shows three grid cells characterized by rather different values of the ratio R of long-term average annual $WU_{sw}$ to long-term average annual $Q_{ant}$: high (Vietnam, 10.75N, 107.25E in Figure 10a), moderate (southeast USA 31.75N, -84.75E in Figure 10b) and low (Russia, 63.75 N, 136.75E in Figure 10c).

As expected, $p_Q$ and *SSFI* show an equivalent behavior in all grid cells as they are based on the same streamflow data, do not use any additional information and can be mathematically transformed from one to the other (Table 1). In contrast, QDAI is based additionally on estimates of the grid cell's specific human surface water demand and assumptions on EFR. A comparison of *SSFI* and QDAI is, therefore, essentially a comparison of $p_Q$ and QDAI. If R is very small, such as in the case of the Russian grid cell, with R = 3.5 x $10^{-6}$ (Figure 10c), QDAI is very similar to $p_Q$, while $d_Q$ is very similar to EFR, being 80% of the mean monthly $Q_{nat}$ (see explanation in Section 3.2). For the Vietnamese grid cell with a high R value of 0.143,

QDAI does not interpret the anomalously low streamflow values in December 2003 and December 2005 as a drought hazard due to the low human water demand for surface water in December. Globally averaged, the fraction of months under drought during 1981-2010 is 16.0% according to QDAI and 19.1% according to SSFI. This reflects that QDAI only identifies a drought condition if there is, in addition to the anomalously low flow, a water deficit.

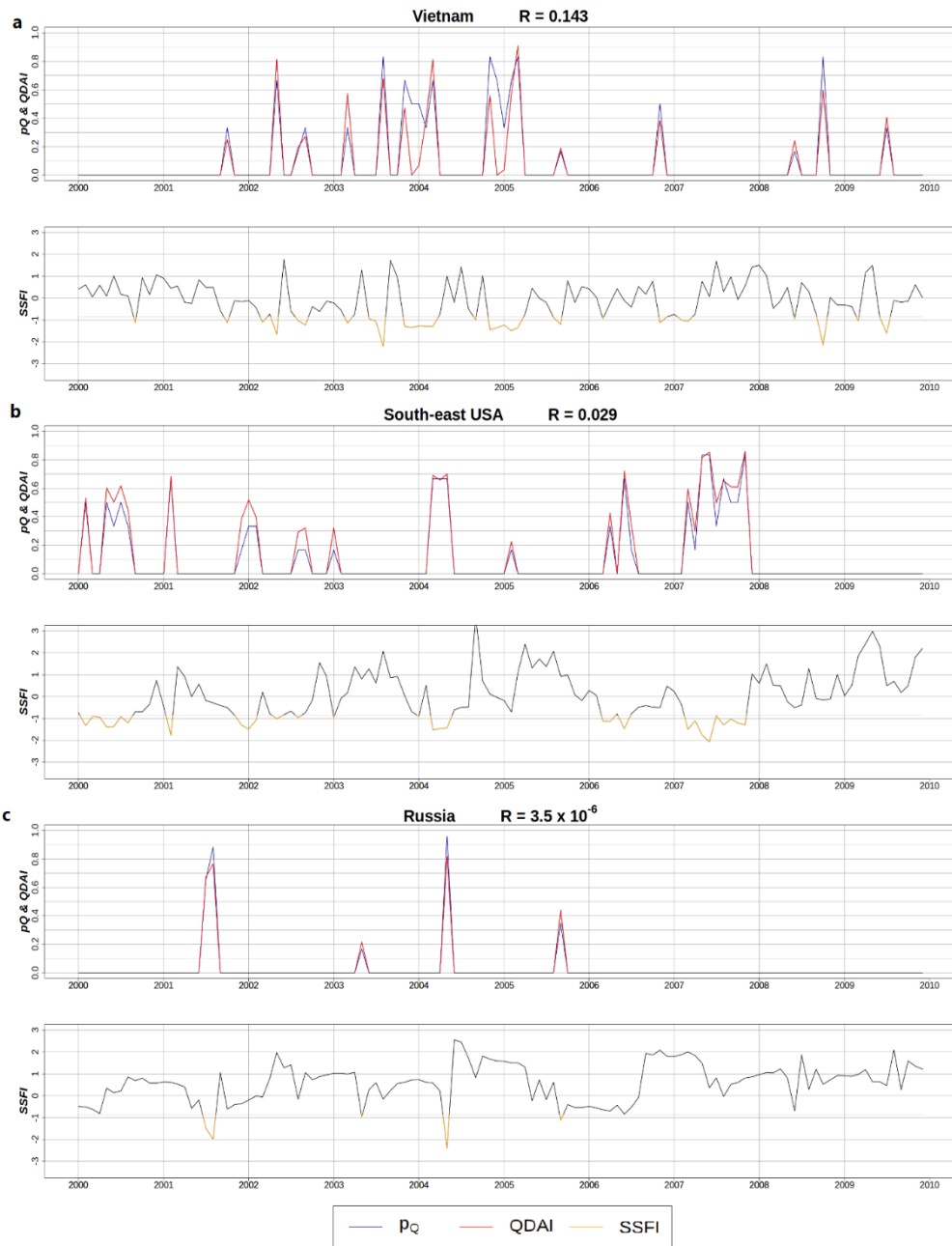


Figure 10. Time series of QDAI and *SSFI* for grid cells with different ratios of surface water abstractions to streamflow R in three regions: a) Vietnam (10.75N, 107.25E), b) south-east USA (31.75N, -84.75E) and c) Russia (63.75 N, 136.75E). *SSFI* is shown in red if it is below -0.84 standard deviations, corresponding to a 5-year return period and a p of zero (Table 1).


**4 Conclusion**

      In this paper, we presented two drought hazard indices that combine the drought deficit and anomaly characteristics: one for soil moisture drought (SMDAI) and the other for streamflow drought (QDAI). With SMDAI, which describes the

drought hazard for vegetation, we achieved the simplification of the deficit-anomaly based Drought Severity Index introduced by Cammalleri et al. (2016). We transferred the DSI concept to streamflow drought, creating an indicator that specifically quantifies the hazard that drought poses for water supply from rivers. To our knowledge, QDAI is the first-ever streamflow drought indicator that combines the anomaly and deficit aspects of streamflow drought.

      The concept of SMDAI and QDAI was tested at the global scale by using simulated data from the latest version of

the global water resources and using the model WaterGAP. Whereas the reliability of the computed SMDAI and QDAI values strongly depends on the quality of the model output. The indicators themselves have been proven to provide meaningful quantitative estimates of drought hazard that depend not only on the unusualness of the situation but also on the concurrent deficit of available water as compared to demand. We found that the values of the combined deficit-anomaly drought indices are often broadly similar to purely anomaly-based indices and share with them the difficulty of dealing with intermittent

streamflow regimes. However, they do provide more differentiated spatial and temporal patterns and help to distinguish the degree and nature of the drought hazard. QDAI can serve as a tool for informing water suppliers and other stakeholders about the joint drought hazard for both water supply for humans and river ecosystem, while stakeholders may adapt the EFR applied for computing QDAI in accordance with their valuation of ecosystem health. Like all hydrological drought indicators that reflect streamflow anomaly, QDAI needs to be interpreted carefully in case of highly intermittent streamflow regimes.

The term "drought hazard" can be defined as the source of a potential adverse effect of an unusual lack of water on humans or ecosystems. In this sense, SMDAI and QDAI are drought hazard indicators, even if they include some elements of vulnerability to drought. Both SMDAI and QDAI are well applicable in drought risk studies. In local drought risk studies, additional indicators of ecological or societal vulnerability should be added, for example, vegetation/crop type or income levels. In regional or global drought risk studies, the inclusion of grid-scale values of QDAI and SMDAI would be beneficial

as both indices contain spatially highly resolved information on vulnerability, while most other vulnerability indicators represent spatial averages of much larger spatial units such as countries.

**Author contributions**

This paper was conceptualized by PD with input from EP. EP performed the data analysis and visualization. The original draft was written by EP and revised by PD.

**Data availability**

WaterGAP 2.2d model output data used in this study are available at https://doi.pangaea.de/10.1594/PANGAEA.918447. The outputs from this study are available at https://doi.org/10.6084/m9.figshare.14213852

**Acknowledgments**

We thank Dr. Hannes Müller Schmied for input and guidance on setting up the WaterGAP variant with doubled $S_{max}$ (Section
4.1.1) and Thedini Asali Peiris for constructive criticism of the manuscript. Also, we acknowledge funding from the German Federal Ministry of Education and Research (BMBF) for the "Globe Drought" project through its funding measure Global Resource Water (GRoW) (grant no. 02WGR1457B).

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
