# Peer review of "Soil moisture and streamflow deficit anomaly index: An approach to quantify drought hazards by combining deficit and anomaly"

_Natural Hazards and Earth System Sciences, 2020_

## Referee Comment (RC1) · Anonymous Referee #1 · 5 Nov 2020

General Comment

The authors introduce two global drought indicators, derived from modeled soisture and streamflow, which are based on the approach proposed by Cammalleri et al. (2016) for soil moisture over Europe. The goal of the study is well presented overall, and the analysis is clear. However, in my opinion, the focus on two indicators make the analysis weaker rather than stronger, and the potentiality of the research is not explored in full. For the first index, the authors introduce some modification to the original formulation of the DSI, but they fail in providing a proof that the proposed simplification is better/equal to the original formulation. For the second, there is much more space

for analysis and discussion on the water demand component of the index, which is a key point of the analysis that is not fully explored. In my opinion, the first part of the analysis is not sufficiently interesting, at least compared to the second, and I suggest to focus solely on the novelty of the streamflow drought and expanding this section for a more efficient delivery of the key message. Overall, I think that the paper has a very good potential, but it needs some major reworks to focus more on the strengths of the research.

Specific comments

Introduction The authors highlights how anomaly-based indicators are usually meteorological indicators, whereas deficit-based indicators are usually soil moisture/evapotranspiration indicators. However, they fail to highlight the reason behind this, which is the difficulty (impossibility?) to define a deficit threshold for a meteorological quantity in absence of a clear target (how much rainfall is enough rainfall?), which is instead more straightforward for plant water demand. This is a key point, an very important for the streamflow drought, where water demand can be defined, but is again much more complex than vegetation demand. I suggest the authors to expand this concept in the introduction to give more impact to the introduction of this concept in streamflow drought.

Methods and data More details on the water demand modules of WaterGAP should be provided, since this is a key component of the streamflow drought index. As stated in the general comment, I would ditch completely the analysis on SMDAI. There is not enough novelty in the modified index as it is, and the introduce modification are not sufficiently tested against the DSI to conclude that this proposed formulation is better/equal to the original (few maps on a specific month of 2003 are not enough). There may be still interest in fully analyzing SMDAI at global scale, since the DSI was tested only over Europe, but this can be the focus of a full expanded paper on this topic, where a detailed intercomparison can be performed. Results Removing the analysis on SMDAI will also give more space to the QDAI, which is more interesting in my opinion

and truly a novelty. However, the analysis is currently lacking in depth in my opinion. As an example, rather than focusing on a single month globally (August 2003), the authors may show some continental maps for different local events. At the moment, its very difficult to draw any consideration of such large maps in which the interesting data are only on a small region. The analysis of the impact of different ecological flow is also very interesting, but it needs to be expanded beyond a couple of months. Finally, the comparison with the SSFI is very useful to give a benchmark to the new index, but it needs to be expanded as well. Is the number of drought event different for different regions? A

---

## Short Comment (SC2) · 5 Nov 2020

This manuscript provide a concrete step ahead in the definition of new and effective soil moisture and streamflow indexes in the context of identification of drought conditions. The authors propose new indexes introducing innovative conceptualization of the soil moisture content also borrowing approaches from a large legacy of consolidated indexes. The outcomes of the manuscript seem convincing and scientifically robust.

---

## Referee Comment (RC2) · Anonymous Referee #2 · 12 Nov 2020

Dear Authors, Dear Editor,

I read with interest this manuscript for its publication in NHESS journal. The work proposes a new formulation for deficit and anomalies indices. Specifically, authors revised the existing Drought Severity Index (DSI) developed by Cammalleri et al., (2016) for soil moisture drought and introduce a new indicator for streamflow drought; both the indicators combine deficit and anomaly aspects of drought.

The paper is well written and organized; the presentation quality is good. The adopted methods are scientifically robust and of interest for the scientific community. I only have some observations and minor comments that can be read in the following.

[Figure]

1. As I understood, distribution functions (gamma and beta) are fitted cell by cell (and for 12 months) all over the globe. Indeed, fir some cells the fits were rejected. Have you thought of carrying out a cluster or regionalization analysis to identify areas with similar parameters thus to improve the fitting? 2. The d_soil component has an almost regular seasonal cycle, as expected. The SMDAI, thus, results to be particularly sensitive to the second component, i.e. the p_soil, which depends on statistical fitting. Clearly, SMDAI is high only when d_soil and p_are are 'in phase', that means, for the case of the German cell in figure 1, when p_soil is high during summer season. This highlights the importance of the fitting process and the utility of possible analyses over regions (previous comment). 3. How along the used monthly time series are? 4. Are there any other recorded drought events against which results can be compared? Summer 2017 was particularly critical for Europe (WWA, 2017). Please discuss.

World Weather Attribution (2017). Euro-Mediterranean Heat - Summer 2017. Report, September 2017

Specific comments L. 165: I would avoid terms such as "unnecessarily complex"; modify in "we have simplified the approach of Cammalleri et al., (2016)". L326-328: please review sentence, something is missing. FIGURES: please, improve quality of map figures. Table 1: specify that the anomaly component p is for both SMDAI and QDAI.

---

## Author Comment (AC1) · 17 Dec 2020

**Answer to comments of Anonymous Referee #1**

The original comments of Referee #1 are in black color and indicated by "R:". Replies by the authors ("A") are colored in green. Actions are introduced by "Action:", changes done in the manuscript are in italics.

**General Comment:**

The authors introduce two global drought indicators, derived from modeled soil moisture and streamflow, which are based on the approach proposed by Cammalleri et al. (2016) for soil moisture over Europe. The goal of the study is well presented overall, and the analysis is clear. However, in my opinion, the focus on two indicators make the analysis weaker rather than stronger, and the potentiality of the research is not explored in full. For the first index, the authors introduce some modification to the original formulation of the DSI, but they fail in providing a proof that the proposed simplification is better/equal to the original formulation. For the second, there is much more space or analysis and discussion on the water demand component of the index, which is a key point of the analysis that is not fully explored. In my opinion, the first part of the analysis is not sufficiently interesting, at least compared to the second, and I suggest to focus solely on the novelty of the streamflow drought and expanding this section fora more efficient delivery of the key message. Overall, I think that the paper has a very good potential, but it needs some major reworks to focus more on the strengths of the research.

A: Thank you for the overall positive comments. We have addressed all your specific comments in the sections below. However, first and foremost, we would like to address here the major suggestion to solely focus on the streamflow drought. Instead of totally removing SMDAI and its analysis altogether from this paper, we believe it is beneficial for the manuscript to relate computed values of SMDAI and QDAI to each other (in addition to just presenting them independently), by analyzing propagation of drought from soil moisture to streamflow.

Action: We have added a new subsection 4.4 which discusses the propagation of drought from soil moisture to streamflow

*"4.4 Propagation of drought from soil moisture to streamflow as indicated by SMDAI and QDAI*

*As can be expected from the flow path of water on the continents, below normal precipitation occurs before below normal soil moisture. Below normal streamflow may occur even later, but only if streamflow at a certain location is not dominated by local conditions and not conditions in a distant upstream area. This so-called drought propagation can be identified by drought hazard indicators for the respective variables (van Loon, (2013). Knowledge about the dynamics of drought propagation supports monitoring drought development and drought mitigation as it allows to estimate, for example, impacts of the early meteorological drought on various sectors at different stages of its propagation through the water cycle. The purely physical propagation may be expected to be best observed by purely*

*anomaly-based indicators, e.g., using standardized drought indicators for the variables: precipitation, soil moisture, and streamflow. Here, we want to explore drought propagation from soil moisture drought to streamflow drought using the deficit-anomaly indicators SMDAI and QDAI.*

*For the example of a grid cell in Germany (42.25N, -121.75 E), drought propagation is identified during the 2003 Central European (CEU) summer drought (Figure 11). Comparing the set of time series for $d_{soil}$, $p_{soil}$, SMDAI with SSFI and $d_Q$, $p_Q$ and QDAI, we observe a lag of one month in the onset of streamflow drought and a two-month delay in the termination of streamflow drought as indicated by QDAI compared to soil moisture drought indicated by SMDAI. Soil moisture drought lasted from March to October 2003, the streamflow drought from April to December 2003. The drought periods by SMDAI and QDAI are driven by their anomaly components $p_{soil}$ and , respectively. However, the highest anomaly of soil moisture is already reached May, and the highest streamflow anomaly only in August. This would indicate a time lag between peak soil and streamflow drought of three months. However, considering SMDAI and QDAI, the time lag is zero, as both peak in August, as soil moisture deficit in March is low. QDAI and $p_Q$ (as well as SSFI) peak in the same month because human water demand in this grid cell is small as compared to the water demand of the ecosystem which is assumed to be a fraction of streamflow. Overall, an extreme soil moisture drought event from June to August 2003 as identified by SMDAI was accompanied and prolonged by a severe streamflow drought event from July to October as identified by QDAI.*

[Figure]

*Figure 11. Drought propagation from soil moisture to streamflow: example of a time series (2002 – 2005) of monthly of $d_{soil}$, $p_{soil}$, $d_Q$, $p_Q$, SMDAI and QDAI for a grid cell in Germany."*

**Specific comments:**

R: Introduction - The authors highlights how anomaly-based indicators are usually meteorological indicators, whereas deficit-based indicators are usually soil moisture/evapotranspiration indicators. However, they fail to highlight the reason behind this, which is the difficulty (impossibility?) to define a deficit threshold for a meteorological quantity in absence of a clear target (how much rainfall is enough rainfall?),which is instead

more straightforward for plant water demand. This is a key point, an very important for the streamflow drought, where water demand can be defined, but is again much more complex than vegetation demand. I suggest the authors to expand this concept in the introduction to give more impact to the introduction of this concept in streamflow drought.

A: We agree with the reviewer that a better formulation of these key points is required.

Action: We replaced the second paragraph of the introduction by

*"        Some researchers have quantified drought by only considering the deficit aspect of drought, i.e., by computing the difference between an optimal water quantity and the actual quantity ("less water than required"). **Deficit-based indicators** have only derived for assessing drought risk for vegetation, as optimal water quantities can be defined by either the field capacity of the soil (Sridhar et al. 2008) or potential evapotranspiration. For the latter, the deficit is computed either as the difference between potential evapotranspiration and precipitation (Hogg et al. 2013)or between potential and actual evapotranspiration. A drawback of these deficit-based drought hazard indicators is that they indicate strong drought in arid and (semi)arid regions, even though the vegetation in these regions is adapted to generally lower soil moisture (Cammalleri et al. 2016). Deficit-based indicators cannot be meaningfully derived for the variable precipitation only as the definition of an optimal precipitation amount depends on the user of the precipitation water. It is, however, conceptually meaningful to determine deficits for human water supply based on the variable streamflow, defining the deficit as the difference between the demand for water from the river and the actual streamflow. To the best of our knowledge, streamflow drought has not, as yet, been characterized by a deficit-based drought indicator."*

R: Methods and data - More details on the water demand modules of WaterGAP should be provided, since this is a key component of the streamflow drought index.

A: We agree with the reviewer.

Action: We replaced the first three sentences of Section 2.1 by the following two paragraphs:

*"        In this study, we use the output of the latest version of the global hydrological and water use model WaterGAP 2.2d (Müller Schmied et al. 2020). WaterGAP consists of three major components: the water use models, the linking model GSWUSE and the global hydrological model (WGHM). The water use models compute water use in the five sectors household, manufacturing, cooling of thermal power plants, livestock and irrigation. Household and manufacturing water use is computed based on national statistics (Flörke et al. 2013). The amount of water required for cooling of thermal power plants is calculated based on location, type and size of power plants and annual time series of thermal electricity production (Flörke et al. 2013). The globally small amount of livestock water use is determined from the number of livestock and livestock-specific water use values (Alcamo et al. 2003). Irrigation water use is computed based on information on irrigated area and climate for each grid cell. The irrigation model first computes cell-specific cropping patterns and growing periods and then irrigation consumptive water use, distinguishing only rice and non-rice crops (Döll and Lehner 2002). The irrigated areas are changing over time (Siebert et al. 2015).*

*The water use models do not take into account the source of the sectoral water abstractions. This is done by GWSWUSE, which computes monthly time series of 0.5° grid-cell values of human water abstractions from 1) surface water bodies (river, lakes and man-made reservoirs) and 2) groundwater, for each of the five sectors, as well as the respective net abstractions from both sources (Döll et al. 2012). A comparison of simulated annual sectoral water abstractions per country to independent values from the AQUASTAT database of FAO showed a rather high similarity between the two data sets (Müller Schmied et al. 2020)."*

R: As stated in the general comment, I would ditch completely the analysis on SMDAI. There is not enough novelty in the modified index as it is, and the introduce modification are not sufficiently tested against the DSI to conclude that this proposed formulation is better/equal to the original (few maps on a specific month of 2003 are not enough). There may be still interest in fully analyzing SMDAI at global scale, since the DSI was tested only over Europe, but this can be the focus of a full expanded paper on this topic, where a detailed inter comparison can be performed.

A: The comment of completely ditching analysis on SMDAI has been addressed before. On the other suggestion, in the paper, we do not state that SMDAI better indicates drought conditions than DSI. We conclude that both result in very similar quantitative drought hazard values while SMDAI is computed in a more straightforward way without the need of introducing an additional mapping equation.

Action: We have modified the paragraph in section 2.2.2 as

*"Cammalleri et al. (2016) calculated $p_{soil}$ using the mode instead of median as the reference for the normal status of $d_{soil}$. The computation of $p_{soil}$ from $F(d_{soil})$ was carried out in two steps. First, for $d_{soil}$ values that are greater than or equal to the mode, a new standardized cumulative distribution function $F*(d_{soil})$ is computed (Eq. 3 in Cammalleri et al., 2016). Subsequently, mapping of $F*(d_{soil})$ values ranging from 0.6 to 1 onto the $p_{soil}$ range of [0, 1], an exponential function (Eq. 4 in Cammalleri et al., 2016) was employed. This exponential function was developed to fit subjectively defined pairs of $F*(d_{soil})$ and $p_{soil}$ (Table 1 in Cammalleri et al., 2016). In this study, we have simplified the more complex approach of Cammalleri et al. (2016) by relying directly on $F(d_{soil})$ for mapping $F(d_{soil})$ onto $p_{soil}$ according to Eq. 3. In our opinion, there is no added value in defining an arbitrary exponential mapping function for deriving an indicator for the probability of a drought occurrence ($p_{soil}$). Further, like most other drought researchers, we prefer the median to the mode, as among 30 deficit values, which are rational numbers, there is no true mode, i.e., no value that occurs most often. The relation between the anomaly component of SMDAI (i.e., $p_{soil}$) to the non-exceedance probability of the soil moisture deficit ($F(d_{soil})$) and the pertaining return periods, z-scores, and class names, according to Agnew (2000) as well as the anomaly component of DSI (p_DSI) are presented in Table 1. A comparison of $p_{soil}$ to p_DSI values as a function of ($F(d_{soil})$) as presented in Table 1 is shown in Figure S1 and the slight differences between $p_{soil}$ and p_DSI, as well as DSI and SMDAI, computed with WaterGAP output for August 2003 at the global scale are presented in Figure S2. For the period 1981-2010, SMDAI is, averaged over all grid cells, 0.05 larger than DSI."*

R: Results - Removing the analysis on SMDAI will also give more space to the QDAI, which is more interesting in my opinion and truly a novelty. However, the analysis is currently lacking in depth in my opinion. As an example, rather than focusing on a single month globally (August 2003), the authors may show some continental maps for different local events. At the moment, its very difficult to draw any consideration of such large maps in which the interesting data are only on a small region. The analysis of the impact of different ecological flow is also very interesting, but it needs to be expanded beyond a couple of months.

A: Comment on removing the analysis on SMDAI has been addressed before. On providing continental maps for different local events as well as more analysis on ecological flow, we agree to the suggestion.

Action: We have added a new paragraph in section 4.2

*"Further differences between QDAI values computed for alternative EFR are explored for two widely known drought events, the South Asian drought of 2009 (Neena et al. 2011) and the North American drought of 2002 (Seager 2007). Figure 9 presents the spatial extent of both the droughts detected by QDAI at a continental scale (left panels of figure 9) for August 2009 and March 2002, respectively. Time series plots (right panels of Figure 9) for an Indian grid cell (75.75 E, 24.75 N top panel), as well as another for a USA grid cell (-110.75 E, 44.25 N bottom panel), provide a better understanding of the sensitivity of QDAI to EFR. As expected, QDAI values calculated with EFR = 0 (green) are lower and drought periods shorter than if it is assumed that water needs to remain in the river for the well-being of the ecosystems. Interestingly, short but severe drought in the Indian grid cell in 2002, 2006, and 2010 have almost equal QDAI values for all three EFR alternatives.*

[Figure]

*Figure 9. Continental maps of QDAI for Asia and Northern America for August 2009 and March 2002 respectively (left panels) with blue points showing the location of the Indian and USA grid cells. Time series of different QDAI with alternative EFR (right panels) for Indian grid cell for 2001-2010 and USA grid cell for 1998 – 2007 and nc are grid cells which are not computed due to land cover"*

R: Finally, the comparison with the SSFI is very useful to give a benchmark to the new index, but it needs to be expanded as well. Is the number of drought event different for different regions?

A: We agree comparison of QDAI with SSFI is a useful addition to the paper. A comparision of the fraction of months under drought conditions at global scale is added.

Action: We have added new lines in section 4.3

*"Globally averaged, the fraction of months under drought during 1981-2010 is 16.0% according to QDAI and 19.1% according to SSFI. This reflects that QDAI only identifies a drought condition if there is, in addition to the anomalously low flow, a water deficit."*

**Reference**

Alcamo, J.; Döll, P.; Henrich, T; Kasper, F; Lehner, B.; Rösch, T.; Siebert S (2003): Development and testing of the WaterGAP 2 global model of water use and availability. In *Hydrological Sciences Journal* 48 (3), pp. 317–337. DOI: 10.1623/hysj.48.3.317.45290.

Cammalleri, Carmelo; Micale, Fabio; Vogt, Jürgen (2016): A novel soil moisture-based drought severity index (DSI) combining water deficit magnitude and frequency. In *Hydrol. Process.* 30 (2), pp. 289–301. DOI: 10.1002/hyp.10578.

Döll, P.; Hoffmann-Dobrev, H.; Portmann, F. T.; Siebert, S.; Eicker, A.; Rodell, M. et al. (2012): Impact of water withdrawals from groundwater and surface water on continental water storage variations. In *Journal of Geodynamics* 59-60, pp. 143–156. DOI: 10.1016/j.jog.2011.05.001.

Döll, P.; Lehner, B. (2002): Validation of a new global 30-min drainage direction map. In *Journal of Hydrology* 258 (1-4), pp. 214–231. DOI: 10.1016/S0022-1694(01)00565-0.

Flörke, M.; Kynast, E.; Bärlund, I.; Eisner, S.; Wimmer, F.; Alcamo, J. (2013): Domestic and industrial water uses of the past 60 years as a mirror of socio-economic development: A global simulation study. In *Global Environmental Change* 23 (1), pp. 144–156. DOI: 10.1016/j.gloenvcha.2012.10.018.

Hogg, E. H.; Barr, A. G.; Black, T. A. (2013): A simple soil moisture index for representing multi-year drought impacts on aspen productivity in the western Canadian interior. In *Agricultural and Forest Meteorology* 178-179, pp. 173–182. DOI: 10.1016/j.agrformet.2013.04.025.

Müller Schmied, H; Cáceres, Denise; Eisner, Stephanie; Flörke, Martina; Niemann, Christoph; Peiris, Thedini Asali et al. (2020): The global water resources and use model WaterGAP v2.2d: Model description and evaluation. In *submitted to Geoscientific Model Development*.

Siebert, Stefan; Kummu, Matti; Porkka, Miina; Döll, Petra; Ramankutty, Navin; Scanlon, Bridget (2015): Historical Irrigation Dataset (HID). With assistance of Lan Zhao.

Sridhar, Venkataramana; Hubbard, Kenneth G.; You, Jinsheng; Hunt, Eric D. (2008): Development of the soil moisture index to quantify agricultural drought and Its "user friendliness" in severity-area-duration assessment. In *J. Hydrometeor.* 9 (4), pp. 660–676. DOI: 10.1175/2007JHM892.1.

---

## Author Comment (AC2) · 17 Dec 2020

**Answer to comments of Anonymous Referee #2**

The original comments of Referee #2 are in black color and indicated by "R:". Replies by the authors ("A") are colored in green. Actions are introduced by "Action:", changes in the manuscript are in italics.

R: Dear Authors, Dear Editor,

I read with interest this manuscript for its publication in NHESS journal. The work proposes a new formulation for deficit and anomalies indices. Specifically, authors revised the existing Drought Severity Index (DSI) developed by Cammalleri et al., (2016) for soil moisture drought and introduce a new indicator for streamflow drought; both the indicators combine deficit and anomaly aspects of drought. The paper is well written and organized; the presentation quality is good. The adopted methods are scientifically robust and of interest for the scientific community.

A: Thank you for the very positive comments and encouragement. All your minor and specific comments have been addressed below.

R: I only have some observations and minor comments that can be read in the following.

1. As I understood, distribution functions (gamma and beta) are fitted cell by cell (and for 12 months) all over the globe. Indeed, fir some cells the fits were rejected. Have you thought of carrying out a cluster or regionalization analysis to identify areas with similar parameters thus to improve the fitting? 2. The d_soil component has an almost regular seasonal cycle, as expected. The SMDAI, thus, results to be particularly sensitive to the second component, i.e. the p_soil, which depends on statistical fitting. Clearly, SMDAI is high only when d_soil and p_are are 'in phase', that means, for the case of the German cell in figure 1, when p_soil is high during summer season. This highlights the importance of the fitting process and the utility of possible analyses over regions (previous comment).

A: Certainly, cluster or regionalization analysis to identify areas with similar parameters could be a good option to have a better distribution fit, at least for SMDAI. However, after checking with 100+ parametric functions and with several different parameter values, we found it is not a very feasible option for these indices. Also, we envisioned to develop both SMDAI and QDAI as more grid-based indices, with the idea that both indices provide highly resolved vulnerability and spatial information. It is expected that QDAI fitting functions in particular cannot be regionalized due to the topology of the river network.

In the paper, for the 27.12% of grid cells in the case of $d_{soil}$ and 39.94% of grid cells in the case of $Q_{ant,}$ that were rejected by the one-sample Kolmogorov–Smirnov test (KS-test at 0.05 significance level), we used the empirical cumulative distribution function (ECDF) to compute the respective non-exceedance values. Please find below a cdf comparing a non-exceedance probability determined by gamma distribution and ECDF for streamflow in a grid cell in central USA for the calendar months of June and December where gamma distribution is not accepted by KS Test (for all 12 calendar months). We think that that in case of an

uninterrupted time series of data, using the simple EDCF approach for deriving a frequency distribution is exceedance probabilities is not necessarily worse that fitting a function.

[Figure]

Figure: examples of CDF plots of the 30 June (left panel) and December values of the reference period (right panel) of streamflow in a grid cell in the Central USA (-99.25E,33.25N), where gamma distribution is not accepted by KS Test (for all 12 calendar months).

3. How along the used monthly time series are?

A: As already pointed out in L.119, we use monthly time series data of 30 years from 1981 to 2010 that are simulation results of the global hydrological model WaterGAP.

04. Are there any other recorded drought events against which results can be compared? Summer 2017 was particularly critical for Europe (WWA, 2017). Please discuss.

A: The climate forcing WFDEI-GPCC, which is standard (and best) input for WaterGAP 2.2d, is only available up to the end of 2016, which is why we cannot analyze more recent droughts. Instead, we have now analyzed other droughts, the South Asian drought of 2009 and the North American drought of 2002 at continental scale, also for showing the sensitivity of QDAI to EFR.

Action: We have added an additional analysis in section 4.2

*"Further differences between QDAI values computed for alternative EFR are explored for two widely known drought events, the South Asian drought of 2009 (Neena et al. 2011) and the North American drought of 2002 (Seager 2007). Figure 9 presents the spatial extent of both the droughts detected by QDAI at a continental scale (left panels of figure 9) for August 2009 and March 2002, respectively. Time series plots (right panels of Figure 9) for an Indian grid cell (75.75 E, 24.75 N top panel), as well as another for a USA grid cell (-110.75 E, 44.25 N bottom panel), provide a better understanding of the sensitivity of QDAI to EFR. As expected, QDAI values calculated with EFR = 0 (green) are lower and drought periods shorter than if it is assumed that water needs to remain in the river for the well-being of the ecosystems.*

*Interestingly, short but severe drought in the Indian grid cell in 2002, 2006, and 2010 have almost equal QDAI values for all three EFR alternatives.*

[Figure]

*Figure 9. Continental maps of QDAI for Asia and Northern America for August 2009 and March 2002 respectively (left panels) with blue points showing the location of the Indian and USA grid cells. Time series of different QDAI with alternative EFR (right panels) for Indian grid cell for 2001-2010 and USA grid cell for 1998 – 2007 and nc are grid cells which are not computed due to land cover "*

Specific comments

R: L. 165: I would avoid terms such as "unnecessarily complex"; modify in "we have simplified the approach of Cammalleri et al., (2016)".

A: Thank you for pointing it out.

Action: L.165 modified to

*"we have simplified the more complex approach of Cammalleri et al. (2016)"*

R: L326-328: please review sentence, something is missing.

A: Thank you for pointing it out.

Action:  We have modified it to

*"In the grid cell in the western USA, where streamflow of the Klamath River is observed in Keno (42.25N, -121.75 E, left panels of Figure 4), water demand is mostly for irrigation (i.e., 0.038 km3 month-1 temporal mean) which is high compared to the relatively small streamflow (i.e., 0.105 km3 month-1 temporal mean)".*

FIGURES: please, improve quality of map figures.

*A: Thank you. We have increased the dpi for a better quality of the figures.*

R: Table 1: specify that the anomaly component p is for both SMDAI and QDAI.

A: The anomaly component p for both SMDAI and QDAI, i.e., $p_{soil}$ and $p_Q$ respectively, are presented as p in the Table 1.

Action: We have modified the heading of the table 1 for improved understanding as follows.

| $F(d_{soil})/$ $F(Q)$ | Return period (yrs) | z-score | Drought class name | p_DSI | $p_{soil}/p_Q$ |
|---|---|---|---|---|---|
|  |  |  |  |  |  |

**References**

Neena, J. M.; Suhas, E.; Goswami, B. N. (2011): Leading role of internal dynamics in the 2009 Indian summer monsoon drought. In *J. Geophys. Res.* 116 (D13). DOI: 10.1029/2010JD015328.

Seager, Richard (2007): The Turn of the Century North American Drought: Global Context, Dynamics, and Past Analogs*. In *J. Climate* 20 (22), pp. 5527–5552. DOI: 10.1175/2007JCLI1529.1.

---

## Author Comment (AC3) · 17 Dec 2020

**Additional changes:**

While working on the valuable referee comments and also by direct feedback from readers of the discussion paper, we further modified the manuscript as follows:

- Added a new section on data availability
- Added interpretation of highly intermittent streamflow regimes with QDAI

---

## Author Comment (AC4) · 17 Dec 2020

Thank you for the overall positive comments
* * *

---

## Author Response (AR2)

**Answer to comments Editor**

The original comments of the editor are in black color and indicated by "R:". Replies by the authors ("A") are colored in green. Actions are introduced by "Action:", changes done in the manuscript are in italics.

**General Comment:**

As you can see from the reviewers' report, there are still some concerns on the paper, that I agree need to be addressed before evaluating the paper for publication. Please revise the paper accordingly, with special attention to the structure of the results and discussion sections and to the need to expand the analysis to other periods relevant for regions outside Europe in order to support the focus on global scale.

-- Thank you for your recommendations. 1) We agree that also that in the discussion section of manuscript, we showed results. We therefore followed the review suggestion and have merged the results and discussion section into one section. 2) Regarding the presentation of global-scale results we already had presented Figs. 3 (SMDAI) and 6 (QDAI), which provided the overall DAI indicator behavior globally and for the whole reference period. Additionally, we had already included the analysis of two non-European streamflow drought events are presented in Fig. 9. We do not think that an additional similar analysis for SMDAI would be suitable for the paper for various reasons: 1) The paper is already very long and more figures would rather overwhelm or bore the readers, 2) The main innovation is in QDAI while the inclusion of SMDAI is done to show that a) both soil and streamflow droughts can be assessed by a deficit-anomaly approach in a parallel and consistent manner and b) the important issue of drought propagation from soil moisture drought to streamflow drought can be analyzed with the introduced approach, 3) any presentation of SMDAI results during non-European drought events would just be illustrative and 4) the manuscript is primarily a methodological study on new drought indicator(s) and not a study on the occurrence of historical droughts around the globe. So we have not included any further analysis results for specific regions and drought periods as for the for reasons given, we strongly fell that this would be detrimental for the paper. Otherwise, we have adjusted the paper in many aspects in response to the helpful comments of the reviewer (please see below).

**Answer to comments of Anonymous Referee #1**

The original comments of Referee #1 are in black color and indicated by "R:". Replies by the authors ("A") are colored in green. Actions are introduced by "Action:", changes done in the manuscript are in italics.

**General Comment:**

I would like to thank the authors for carefully considering my comments and revisiting the manuscript accordingly. Even if I found the new version significantly improved compared to the original version, I still have some major concerns on some details of the methodology, as well as on the way the results are presented, especially in the framework of a work on global scale drought.The authors introduced some simplifications from the original formulation of the DSI, but in my opinion they fail in highlighting if/how these simplifications impact on the behavior of the index.

-- In Lines 182-85, the comparison between both $p_{soil}$ and p_DSI is mentioned along with the slight difference between the two for Aug 2003 observed in respective world maps represented in FigS2.

Action : To be more precise, we have added the following in line 184:
*"For very few gridcells, SMDAI is much larger than DSI and there are some areas where DSI is slightly larger than SMDAI."*

In my understanding, the two main differences are: 1) the d index is compute on the Smax, and 2) the computation of p is performed differently. From the reported results I see a very high fraction of data with high d values (Fig. 2), and I am wondering how much of this is related to the adopted modification.

-- In our global scale model, different from European scale model used in Cammalleri et al (2016), we do not have the information on water content at wilting point and field capacity, which is why we cannot compute the deficit according to Eq. 1 of Cammalleri et al. for the global scale. With our definition of d (Eq. 1 in our manuscript), computed d at small soil moisture saturation values are smaller than with the definition of Cammalleri et al (2016). (their Eq. 1 and their Fig. 2 ) while d at high soil moisture saturation is not as close. This is due to the S-shaped d-curve of Cammalleri, while our d-curve is a straight line. We do not know whether Camalleri et al. compute smaller d values for August 2003. Anyway, differences would also be caused by the different hydrological models used to compute soil moisture in the two studies. Following the reviewers suggestion below, we added text to section 2.1.1 on how the deficit definitions differ between our approach and the approach of Cammalleri et al (2016).

In term of p, the new Figure S1 show to me a quite different behavior compared to DSI (e.g. for F = 0.87 P is 0.1 and 0.4, respectively).Similarly, Fig. 1 is supposed to show two contrasting examples, as done in Cammalleri et al. for DSI. However, these two cases do not seem opposite example at first glance. If you look at Fig. 3 in the DSI paper, in one case the DSI resembles d while in the other resembles p. In your analogous figure, both cases resemble p. This means that either you selected two

cases that are too similar or that your simplifications do a disservice to the index. Please provide better examples or clarify.

When we evaluated Norwegian grid cell shown in Fig. 3 of Cammalleri et al (2016), we found that with our deficit approach and the output of our global hydrological model, deficits were not zero (or very close to zero), different from the results of Cammalleri et al. We think this is both due to our deficit definition (see explanation of differences above) and the applied hydrological model. We think that the Cammalleri deficit equation, where deficit is not a linear function of soil moisture saturation, does lead to more distinct deficit/no-deficit identifications. We do not think that our d definition is better than the one of Cammalleri et al (2016) and do not express this in the paper, but we also think that it is not necessarily worse. For our Fig. 1, we selected a grid cell in India that different from the grid cell in Spain has a very low deficit most of the time but different from the Norwegian cell in Cammalleri et al (2016) there are longer periods with a small (but non-zero) deficit. Therefore, different from the Norwegian cell in Cammalleri et al (2016), p peaks do not completely vanish. However, we do see a stark contrast between the Spanish and the Indian cell, as in the Indian cell the SMDAI is always much smaller than the anomaly p_soil.

Following my main concern in the first round of review, I still found the section on results quite lacking in the context of a global study. Too much emphasis is given to a specific arbitrarily-selected month (August 2003), and the addition of few figures in the supplementary materials does not alleviate the issue. I strongly suggest to the authors to reshape the approach adopted to show the results, in a way that better highlight the results during relevant droughts. As you stated, most of the globe is likely to be in no drought during any given month, so it is meaningless to show the behavior of the index during such period.I much prefer the approach adopted in Fig. 9 for the EFR, and I suggest to expand this event-based approach to the other analyses as well. This is valid for both indices.

We have selected August 2003 for as an example for showing global maps of SMDAI (Fig. 2) and QDAI (Fig. 5) because it was a month with an extreme drought in Central Europe, to which we refer also in the time series plots for SMDAI (Figs. 1) and QDAI (Fig. 4). To cover more than one arbitrary month but show the overall behavior over the whole time series and globally, we prepared Figs. 3 (SMDAI) and 6 (QDAI). We disagree with the reviewer that it is meaningless to show where there is no drought.

We have already included the analysis of two non-European drought events regarding 9) and do not think that an additional similar analysis for SMDAI would be suitable for the paper for various reasons. 1) The paper is already very long and more figures would rather overwhelm or bore the readers, 2) The main innovation is in QDAI while the inclusion of SMDAI is done to show that a) both soil and streamflow droughts can be assessed by a deficit-anomaly approach in a parallel and consistent manner and b) the important issue of drought propagation from soil moisture drought to streamflow drought can be analyzed with the introduced approach, 3) any presentation of SMDAI results during non-European drought events would just be illustrative and 4) the manuscript is primarily a methodological study on new drought indicator(s) and not a study on the occurrence of historical droughts around the globe.

Also, at the moment there is no clear distinction between the "results" and the "discussion" section, with both containing what can be called results. The discussion

section should discuss the outcomes reported in the results section, not adding new outcomes. Please improve this structure, even by simply merging the two into a single results and discussion section and homogenize.

-- Thank you for your recommendation. We have merged the results and discussion section.

Overall, I see a lot of potentiality in this paper, and a large amount of valuable data that can really help the drought community in improving the understanding of both soil moisture and river droughts, but I think that these data need to be better presented to the readers to transfer the right message.

Specific Comments

L8-9. remove "the condition of"
---- Action: Thank you for pointing it out. We have adapted the suggested changes.

L11. Please add full reference to the DSI in the abstract.
--- During the initial submission, the editor had advised that it would be better to avoid references in the abstract.

L12. Too many details for an abstract. Please reword "… is based on… mapping function" as something in the line "…as a simplified version of the DSI (ref)".

--- Thank you for your suggestion. However, the term simplified would not reflect our modification. We feel that it is important to indicate it already in the abstract. What the implemented improvement consists in.

L39. I think that "can be used" is not necessary here.
---- Action: Thank you for pointing it out. We have adapted the suggested changes.

L98. outputs.
---- Action: Thank you for pointing it out. We have adapted the suggested changes.

L100. Please add ":" after "five sectors".

---- It is already there L100: *"The water use models compute water use in the five sectors: household, manufacturing, cooling of thermal power plants, livestock and irrigation"*

L109-110. I suggest to reword this sentence. At first glance, it seems that "source of water abstraction" is not accounted, and not that this is done by a different module.

--- Action: The new sentence now starts with *" The water use models themselves …"* instead of *"The water use models ..."*

L131-133. It is important to mention specific validations made on low-flow/drought (if any). It is well-known that performances on the lower spectrum of the flow regime may differ quite significantly from the average/high regimes. In absence of specific validations on drought, it is worth mentioning this other possible source of error.

--- Action: We added the following sentence in line 135:

*"It is found that WaterGAP can simulate the low flow percentile (Q95) very well, but it can also overestimate the return period of low streamflow (Zaherpour et al., 2018). "*

L135-136. This sentence needs some rewording in my opinion.
---- Action : We have replaced:

*"This study uses simulated data of 30-years (1981 – 2010)  monthly time series of WaterGAP gridded (0.5° x 0.5° ) output of 67420 land grid cells covering all land areas of the globe except Greenland and Antarctica, for 1) soil moisture $(S)$ [mm], 2) streamflow $(Q_{ant})$ [km3 month-1], 3) streamflow under naturalized condition $(Q_{nat})$ [km3 month-1], assuming there are no human water abstraction or man-made reservoirs, and 4) total surface water abstractions $(WU_{sw})$ [km3 month-1]."*

*By*

*This study uses 30-years (1981- 2010) monthly time series of WaterGAP gridded (0.5° x 0.5°) outputs for 67420 land grid cells covering all land areas of globe except Greenland and Antarctica. These include 1) soil moisture  [mm], 2) streamflow [km3 month-1], 3) streamflow under naturalized condition  [km3 month-1], assuming there are no human water abstraction or man-made reservoirs, and 4) total surface water abstractions ) [km3 month-1]."*

L144. Here I see a major difference between this approach and the one proposed in DSI. If I understand correctly, in DSI the critical water content is used (50% of filed capacity) rather than the field capacity itself. Indeed, the absence of water stress starts in many cases well before field capacity in reached. I suggest to make this difference much more clear (see also discussion 4.1) and to clarify the impact on the high d values observed over most of the globe in fig. 1.

---- We thank you for this suggestion.

Action: We have added the following text to section 2.2.1 where we introduce the computation of the soil moisture deficit d.

*"This definition of soil moisture deficit is different from the definition used in Cammalleri et al. (2016, their Eq. 1) because their definition cannot be applied when using the global hydrological model WaterGAP to compute soil moisture. The deficit computation according to Cammalleri et al. (2016) requires data on soil moisture content at wilting point and at field capacity, which is not available in WaterGAP. With our approach, which is consistent with the way of computing actual evapotranspiration from potential evapotranspiration in WaterGAP, d-values at low soil moisture*

*saturation are lower than those of Cammaleri et al. (2016), while at high saturation they are higher. Consequently, we identify very few months and grid cells with a deficit of zero, likely less than we would do if we would have implemented the deficit definition of Camalleri et al. (2016)."*

L211-216. This is a quite key point, that needs to be stressed more. It would be interesting to have some more insight on the differences with anomalies computed on the deficit, even if no detailed analysis is provided.

---Thank you for the suggestion.

Action: We replaced the sentence

*"The unusualness of a streamflow drought is better captured by a standard cumulative distribution function that can reproduce the statistical structure of streamflow (Qant) compared to a standard distribution function reproducing the statistical structure of streamflow deficit (dQ) due to the temporal variability of the water demand."*

*by*

*"We select to consider the anomaly of streamflow (Qant) instead of the anomaly of the streamflow deficit (dQ) as the temporal variability including long-term trends of the water demand prevented us, for most grid cells with relevant water demand, from identifying a standard distribution function for the time series of dQ."*

The revised sentence gives some insights into the anomalies of streamflow vs. anomalies of streamflow deficit and explains why we could not consider the anomaly of the deficit.

L245. It would be more consistent to use the ECDF for all the cells, especially since you are not using the mode as reference (which needs to be derived from the theoretical distribution), even if I guess that there is not much difference in the case of a good fitting. I also suggest to integrate figure S4 here rather than as supplementary material.

--- We think that it is better to fit optimal functions where it is possible (as is done in most drought studies). And we agree with you that it will not change much the results in those grid cells where we could identify CDFs. We think that we should keep Fig. S4 in the supplementary material, as it provides just a background explanation and not a central methodological information or a study result.

L252. The relationship between dsoil and what? Please rephrase.

--- Thank you. Action: We change the sentence as:

*" The relations between $d_{soil}$, mean monthly ($d_{soil\_mean}$), $p_{soil}$ and SMDAI are further clarified by time series of these variables in Figure 1 for two grid cells with rather different characteristics: a grid cell in Germany (42.25N, -121.75 E, left panels in Figure 1) and one in northeast India (88.25 E,27.25 N, right panels in Figure 1)."*

L264. "very high soil moisture saturation". Please reword.

-- Unfortunately, we cannot think of a better term.

L286. Still, very surprising that most of Canada during December 1999 is in high water stress. The high fraction of the world with dsoil > 0.75 is very surprising in general. This need a clear explanation (it seems to happen also in December 1999). Also, I do not think that the analysis of a single month (out of 30+ years) is enough here. You need to come up with a synthetic map that summarize the whole period (or the major events for different areas), not just a single case (see major comments).

---The low soil moisture in Canada in winter is due to sub-zero temperatures, such that all precipitation falls as snow and does infiltrate the soil. In WaterGAP, like in most hydrological models, soil freezing and permafrost is not taken into account, and in case of no liquid water entering from the top of the soil, the soil drains downward and becomes more and more unsaturated.

Action : We added in  the following text in italics in line 286:

"but high in most snow-dominated northern high-latitude regions *(as no liquid water enters the soil)*,

Regarding a summary presentation over the whole 30 years, we have done this in Figs. 3 and 6 for SMDAI and QDAI, respectively.

L301-302. Is it realistic to call this soil moisture drought, when most of the deficit is due to snow fall rather than liquid precipitation (hence, the soil is covered in snow)? I see this more of a problem for the indicator rather than a desired behavior. Example: How is a good thing that Australia and Canada behave the same?

--- We expect that trees in Canada during months with below zero temperatures do suffer from some stress from low water availability. While it may be true that they suffer more from the low temperatures than from the low soil water saturation, any vegetation during cooler winters, with e.g. temperature below 5-10 °C, will react to lower temperatures, too. We believe that it is beyond the scope of a drought study to take the combination of temperature and water stress into account.

L308-310. this is an unnecessary introduction.

--- We prefer to keep this sentence to better introduce the readers with respect to the results presented in the following sentences.

L337. The effect of EFR in defining a drought is quite important and needs to be better highlighted in my opinion. At the moment, relegating this analysis to a supplementary material does not give justice to a really key point of transferring this concept from soil moisture to stream flow.

--- Thank you for the comment. We agree that EFR is important for defining streamflow and hence, we have already committed an entire subsection (4.2) on it. With 11 figures, with main text is already unusually high for a research paper

L337-340. I am not following this argument. It seems to me that QDAI is rather similar to p in the cell over US, and I do not see any major clear differences between the two sites in term of "strength" of the droughts. Please elaborate better this concept.

Action: Thank you for your comment. We have adapted the required the required sentences ( L 331 – 346)

" *Characterized by a high seasonality, anthropogenic surface water demand,$WU_{sw}$ (dashed grey line in center plot) and total surface water demand (i.e., $WU_{sw} + EFR\_0.8$, orange line in center plot) result in very high deficits $d_Q$ (green line of the bottom plot) during almost every summer. However, there are only a few months with drought as identified by the anomaly-based drought hazard $p_Q$ exceeding zero (dark blue line).This occurs because the decade shown in Figure 4 happens to be a very wet decade compared to the whole reference period. Another reason is that more than 20% of the years show zero streamflow in the calendar months August and September such that $p_Q$ is zero in all 30 August and September months of the reference period, i.e. no drought is indicated even in case of zero streamflow (see left panel of Figure S7). Due to the large deficit values, $p_Q$ is almost always smaller than $d_Q$ in this US grid cell.*

*In the German grid cell (right panels in Figure 4), the relatively low anthropogenic surface water abstractions result in almost identical values of $Q_{nat\_mean}$ and $Q_{ant\_mean}$ (lines overlap in the top plot), and total surface water demand is very similar to $EFR$ (lines overlap in the center plot). Non-zero $d_Q$ values (bottom plot) are mainly computed if $Q_{ant}$ is lower than $EFR$, such as during the central European drought of 2003. It is sensible to consider this type of situation as a drought hazard as water supply companies would have to stop any surface water abstraction if they wished to protect the river ecosystem. Different from the US grid cell, droughts are rather equally distributed over all decades of the reference period in the German grid cell but the summers of 2003 and 2005 suffer from the most severe droughts of the reference period, in line with expected dryer summer due of climate change. Even if taking into account EFR as 80% of of $Q_{nat\_mean}$ ($EFR_{0.8}$), the total surface water demand is so low that in contrast to the US cell, $d_Q$ is always smaller than $p_Q$.*

*Assumptions about the magnitude of EFR have a strong impact on $d_Q$ and thus QDAI of all grid cells except those with very high surface water abstractions such as the US cell. If the water demand of the ecosystem were assumed to be only 20% of $Q_{nat\_mean}$ ($EFR\_0.2$ ) instead of 80% of $Q_{nat\_mean}$, $d_Q$ decreases somewhat in the US cell but reduces to zero during the whole reference period in the German cell (Figure S6). Therefore, water suppliers in the German grid cell would not suffer from any drought hazard (as indicated by QDAI) and would not have to decrease their surface*

*water abstractions even during a drought similar to the 2003 central European drought."*

L352. "…is mostly smaller than less than…"

Action: Thank you for pointing it out.

We have changed it to *".... is mostly less than..."*

L354. Cells with recurring zero-flow should be treaded differently, since the deficit (how low is Q compared to the historical data) cannot be used as reliable quantity for drought. The length of dry spells is considered a much better proxy here. I suggest to better clarify that such areas need to be masked from the analysis (as successively discussed in L365-366). Please also use a different color for these areas, since the current color is too similar to the dark red used for extreme drought (it is the case also for Fig. 6, see int vs. high frequency).

-- In Figure 5, when analyzing a QDAI in a specific month, it is not necessary to exclude cells with more than 6 months (of the 30 months for each calendar months) as the ECDF will be indicative of the anomaly p (see Fig. S7 left). This is different from Fig. 6 where we also show frequencies of no-drought conditions. We followed your advice and changed the color of the masked out cells in Figs. 5 and 6.

L361. "…no-drought conditions according to QDAI (Figure 6)…"

---- Action: Thank you for pointing it out. We have adapted the suggested changes.

L399. A clear definition of (semi)arid and humid is needed in the methodology. Also, what about the other climates? Where all the cells classified as either of the two, and how?

---- Action: Thank you for your recommendation. We have added the definition of (semi)arid and humid in supplement.

L407. I do not really see any major differences between the two months, which is also kind of expected if you do a global average. Differences can be related only to the fact that there is more land in the northern hemisphere compared to the southern. Again, I see more useful an analysis on the full dataset (or specific drought events) rather than 2 randomly selected months that give very similar outcomes.

–Action: We have added an additional box-plot in Fig 8. where global distribution of QDAI in August 2003 (left), December 2003 (middle) and for all 360 months of the reference period (right), computed with alternative assumptions about **EFR** for grid

cells with humid and (semi)arid conditions. Grid cells where all three **EFR** assumptions result in QDAI = 0 are not included.

L449. These two examples need to be better highlighted in Figure 10. For what I can see, even SSFI has no drought at the end of 2005, or maybe you are referring to end of 2004. What I see is two drought periods in SSFI across the 2004 and 2005 lines. Also, what about the other two cases? No examples where QDAI improves on SSFI?

-- Thank you for your comment. We have already addressed the following in L445 - 452

4.4 I miss the role of this section, which again focuses only on a single point and a single case. Even on the specific case, what is the message that you are trying to pass here on this known phenomenon?

-- We want to suggest that two proposed indicators can be used together to analyse drought propagation.

Fig. 4. There are some inconsistencies in the plots. The legend of the plots on the bottom line seems to be off, also according to the text (d should be in green and p in blue).

Action: Thank you. The required corrections have been made in figure 4.

Also, in the top-left panel Qant seems to be always below Qant_mean, which suggest to me that the opposite occurs before 2000. This can be related to a trend in the data, which should be highlighted and discussed.

Action: Thank you for the comment. We observed that the decade shown in Figure 4 happens to be a very wet decade compared to the whole reference period have adapted the required paragraph (L: 331 – 346) which discusses and highlights the same as follows:

*" Characterized by a high seasonality, anthropogenic surface water demand, $WU_{sw}$ (dashed grey line in center plot) and total surface water demand (i.e., $WU_{sw} + EFR\_0.8$, orange line in center plot) result in very high deficits $d_Q$ (green line of the bottom plot) during almost every summer. However, there are only a few months with drought as identified by the anomaly-based drought hazard $p_Q$ exceeding zero (dark blue line).This occurs because the decade shown in Figure 4 happens to be a very wet decade compared to the whole reference period. Another reason is that more than 20% of the years show zero streamflow in the calendar months August and September such that $p_Q$ is zero in all 30 August and September months of the reference period, i.e. no drought is indicated even in case of zero streamflow (see left panel of Figure S7). Due to the large deficit values, $p_Q$ is almost always smaller than $d_Q$ in this US grid cell.*

*In the German grid cell (right panels in Figure 4), the relatively low anthropogenic surface water abstractions result in almost identical values of $Q_{nat\_mean}$ and $Q_{ant\_mean}$ (lines overlap in the top plot), and total surface water demand is very similar to $EFR$ (lines overlap in the center plot). Non-zero $d_Q$ values (bottom plot) are mainly computed if $Q_{ant}$ is lower than $EFR$, such as during the central European drought of 2003. It is sensible to consider this type of situation as a drought hazard as water supply companies would have to stop any surface water abstraction if they wished to protect the river ecosystem. Different from the US grid cell, droughts are rather equally distributed over all decades of the reference period in the German grid cell but the summers of 2003 and 2005 suffer from the most severe droughts of the reference period, in line with expected dryer summer due of climate change. Even if taking into account EFR as 80% of of $Q_{nat\_mean}$ ($EFR_{0.8}$), the total surface water demand is so low that in contrast to the US cell, $d_Q$ is always smaller than $p_Q$.*

*Assumptions about the magnitude of EFR have a strong impact on $d_Q$ and thus QDAI of all grid cells except those with very high surface water abstractions such as the US cell. If the water demand of the ecosystem were assumed to be only 20% of $Q_{nat\_mean}$ ($EFR\_0.2$) instead of 80% of $Q_{nat\_mean}$, $d_Q$ decreases somewhat in the US cell but reduces to zero during the whole reference period in the German cell (Figure S6). Therefore, water suppliers in the German grid cell would not suffer from any drought hazard (as indicated by QDAI) and would not have to decrease their surface water abstractions even during a drought similar to the 2003 central European drought."*

---

## Author Response (AR3)

**Answer to comments Editor**

The original comments of the editor are in black color and indicated by "R:". Replies by the authors ("A") are colored in green. Actions are introduced by "Action:", changes done in the manuscript are in italics.

**Comment:**

Dear Authors

The reviewers are overall satisfied by your replies. Please include the following minor corrections to the manuscript: I will consider the publication after these comments are integrated.

L11. Please add reference to the DSI. The abstract should stand independently of the rest of the text (see guideline of NHESS).

--- Action: Thank you for your recommendation, we have done the required changes.

L25 (BoN, 2018) should be outside the quotation marks.

--- Action: Thank you for your recommendation, we have done the required changes.

L55-56. "…it indicates…(Dai et al., 2004)". Please reword the sentence, which is not well structured at the moment.

--- Action: Thank you for your recommendation, we have rephrased the sentence as follows.

*"Its strengths and weakness  have been well investigated by Dai et al. (2004) and is extensively used in the USA to indicate meteorological droughts (Heim, 2002)."*

L72. Differently…

--- Action: Thank you for your recommendation, we have done the required changes.

L80. (That which should…) Please rephrase.

--- Action: Thank you for your recommendation, we have rephrased it to.

*"In the soil moisture deficit anomaly index (SMDAI), the deficit is calculated as the difference between the soil moisture at field capacity ( which allows optimal and non-water-limited plant growth) and the actual soil moisture."*

L91. I suggest to invert (a) and (b), since this is the order in which the two sub-section are then discussed successively.

--- Action: Thank you for your recommendation, we have done the required changes.

L98-108. Please clarify which component is "static", if any (i.e. do not account for inter-annual or intra-annual fluctuations).

--- Action: Thank you for your recommendation, we have done the required changes.

In L109: *"The globally small amount of livestock water use is the only temporally constant water use and is determined from the number of livestock and livestock-specific water use values (Alcamo et al., 2003). Water use for households, manufacturing and cooling of thermal power plants are constant throughout the year but change from year to year."*

L149. Cammalleri et al. (2016) scale soil moisture between "wilting point" and "critical point", not saturation. Please clarify.

--- For DSI (Cammalleri et al.,2016), the soil moisture deficit is larger than zero only if soil moisture drops below 50% of field capacity. However, in SMDAI, this is the case only if the soil moisture deficit is larger than zero. Further, we have further adapted the following sentence for more clarity:

*L154: "With our approach, which is consistent with the way of computing actual evapotranspiration from potential evapotranspiration in WaterGAP, d-values at low soil moisture saturation are lower than those of Cammalleri et al. (2016), while they are much higher at high soil moisture as Cammalleri et al. (2016) assume that deficits only occur if soil moisture is less than 50% of field capacity."*

L151. Please remove "saturation".

--- Action: Thank you for your recommendation, we have done the required changes.

L153. "…while at high saturation they are higher". This is surprising to me, since Cammalleri et al. define d=1 at critical point, which is much lower than saturation. I would expect to have much less d=1 in you approach as well (similar to zeros). Please clarify.

--- Action: For clarification, we have modified the sentences as follows:

*L154: " With our approach, which is consistent with the way of computing actual evapotranspiration from potential evapotranspiration in WaterGAP, d-values at low soil moisture are lower than those of Cammalleri et al. (2016), while they are much higher at high soil moisture as Cammalleri et al. (2016) assume that deficits only occur if soil moisture is less than 50% of field capacity."*

L211-L214. Please simplify and remove eq. (5). This can read as simple as "EFR is calculated for each calendar month as80% of mean monthly streamflow…, assuming…"

--- Action: Thank you for your recommendation, we have rephrased it to

*"Following Richter et al. (2012), $EFR$ is calculated for each calendar month as 80% of mean monthly streamflow under the naturalized condition ($\overline{Q_{nat}}$), assuming that 80% of the natural mean monthly streamflow that would have occurred in the river without human water use and man-made reservoirs needs to remain in the river for the well-being of the river ecosystem"*

L258. Please add few lines specifying the locations where the ECDF is used (i.e. Australia, south Africa, etc.). Also, looking at figure S4, it seams to me that ECDF is used in more than 1/3 of the areas. Please check the values reported in the text.

--- Thank you for your suggestion. We checked the values reported in the text, the individual percentages (27.12 for SMDAI and 39.94% for QDAI) are correct. The percentages are computed for 57043 grid cells were considered in this study.

L274. Please remove "saturation". You can use soil water content instead of soil moisture.

--- Action: Thank you for your recommendation, we have done the required changes.

L283. can be further explored.

--- Action: Thank you for your recommendation, we have done the required changes.

L284. This high percentage needs to be discussed. Possible reasons? It is also worth to highlight that SMDAI resemble psoil in this case only because dsoil is quite high everywhere. This is not always the case.

--- For SMDAI, the soil moisture deficit is larger than zero if soil moisture drops below field capacity while for DSI (Cammalleri et al.,2016). this is the case only if soil moisture drops below 50% of field capacity. SMDAI resembles psoil if dsoil is similar to psoil. With a relatively high dsoil as compared to the DSI approach, SMDAI is just more likely to be larger than psoil as compared to the DSI approach. We have already highlighted that dsoil in our approach is likely to be higher than DSI approach.
L 154: *"Consequently, we identify very few months and grid cells with a deficit of zero, likely less than we would do if we would have implemented the deficit definition of Cammalleri et al. (2016)."*

L321. "…on more individual variable…". Please add "(i.e…..)". I thin it is worth a further reminder for the readers on the variables that play a role.

---Action: Thank you for your recommendation, we have adapted the sentence accordingly.

*"QDAI depends on more individual variables (i.e., $Q_{ant}$, $WU_{sw}$ and $EFR$) than SMDAI (i.e. S and $S_{max}$)."*

L342. I suggest to replace "sensible" to 'reasonable".

--- Action: Thank you for your recommendation, we have done the required changes.

L348-L353. This paragraph seems a little out of place to me, since a dedicated section to this topic is successively presented. I suggest to shrink this paragraph, and reference the successive analyses.

--- Thank you for your suggestion. We prefer to keep the paragraph here because it refers directly to the two example grid cells discussed above. While in section 3.4 we do a global analysis of QDAI to different assumptions about EFR

L378. "if for many….". Please check this sentence, which is currently unclear.

--- Action: Thank you for your recommendation, we have done the following changes:

*"Besides, grid cells with intermittent flows also show a high percentage of no-drought conditions, if for any calendar month there are at least six months (i.e., at least 20% of the months) with Qant = 0 (Figure S7)."*

*to*

*"Besides, grid cells with intermittent flows also show a high percentage of no-drought conditions, as for any calendar month with at least six months without streamflow pQ is always equal to zero (Figure S7)."*

L396-397. Please rephrase and remove the incorrect term "more amount of soil moisture…".

--- Action: Thank you for your recommendation, we have done the following changes:

*"With Smax2, more amount of soil moisture is kept in the soil and soil deficits, expressed relative to Smax, can be observed to increase or decrease with doubled Smax (Figure 7b)."*

*to*

*"With doubled $S_{max}$ , mean monthly soil moisture increases, too. In most grid cells, the soil moisture deficit increases as compared to standard $S_{max}$ (Figure 7b)."*

L397-398. "Differences are mostly small…". Is this a sign that d is not that sensitive the absolute changes in Smax? So, for higher Smax also S will be higher? If this is the case, I think it is worth to be highlighted.

--- Action: Thank you for your recommendation, we have highlighted it in the following sentence.

*L412: "With doubled $S_{max}$ , mean monthly soil moisture increases, too. In most grid cells, the soil moisture deficit increases as compared to standard $S_{max}$ (Figure 7b)."*

L408. I would say "defined" rather than "computed".

--- Action: Thank you for your recommendation, we have done the required changes.

L445-455. This description needs to be moved to the methodology section. Here you should report only the result of the comparison, but not the description of the SSFI.

--- Action: Thank you for your recommendation. We have moved the description to a new section 2.6 in methodology.

L474. As already highlighted in the previous review rounds, this analysis is a little lacking and not fundamental for the goal of the paper. I suggest to remove this section, and correct the text (e.g. abstract) accordingly.

--- Action: Thank you for your recommendation. We have removed section 3.6. There was no need to adapt any other text.

L508. outputs.

--- Action: Thank you for your recommendation, we have done the required changes.

L509. …but also on the concurrent deficit….

--- Action: Thank you for your recommendation, we have done the required changes.

L520. Please expand a little on these "additional indicators". Are you referring to more detailed information on water use?

--- Action: Thank you for your comment, we have adapted the sentence as follows:

*"In local drought risk studies, additional indicators of ecological or societal vulnerability should be added, for example vegetation/crop type or income levels."*

L528. data are available. Please, also add a sentence of the availability of the outputs of this study (SMDAI and QDAI).

--- Action: Thank you for your comment we have added a sentence on the availability of SMDAI, QDAI and other analysis outputs

*"The outputs from this study are available at https://doi.org/10.6084/m9.figshare.14213852 "*

---

## Author Response (AR4)

**Answer to comments Editor**

The original comments of the editor are in black color and indicated by "R:". Replies by the authors ("A") are colored in green. Actions are introduced by "Action:", changes done in the manuscript are in italics.

**Comment:**

Dear Authors,
I'm glad to inform you that the paper is accepted for publication in our special issue. The publication is subject to some minor technical corrections here listed:

- references in the abstract should be avoided. If their inclusion is needed for a better understanding of the text (as in your case), you should add the full reference (i.e. journal, number , pages) in addition to the authors and year (abstract should be readable as stand-alone text).
- please check the references to eqs. throughout the text. The order of the eqs. has been changed in the latest version, but some references seem to be still based on the old numbering.

--- Action:

Thank you for accepting our paper for publication.

As per your suggestion, we have added the full reference in the abstract
L11: *"The soil moisture deficit anomaly index, SMDAI, is based on the drought severity index, DSI (Cammalleri et al., 2016, Hydrol. Process., 30, 289–301), but is computed in a more straightforward way that does not require the definition of a mapping function."*

Also, have checked for the references to eqs. and done the required changes.